# Class B1 GPCR activation by an intracellular agonist

Kazuhiro Kobayashi[1,7], Kouki Kawakami[2,7], Tsukasa Kusakizako[1], Atsuhiro Tomita[1,6], Michihiro Nishimura[1], Kazuhiro Sawada[1], Hiroyuki H. Okamoto[1], Suzune Hiratsuka[2], Gaku Nakamura[2], Riku Kuwabara[2], Hiroshi Noda[3], Hiroyasu Muramatsu[3], Masaru Shimizu[3], Tomohiko Taguchi[4], Asuka Inoue[2✉], Takeshi Murata[5✉] & Osamu Nureki[1✉]

G protein-coupled receptors (GPCRs) generally accommodate specific ligands in the orthosteric-binding pockets. Ligand binding triggers a receptor allosteric conformational change that leads to the activation of intracellular transducers, G proteins and β-arrestins. Because these signals often induce adverse effects, the selective activation mechanism for each transducer must be elucidated. Thus, many orthosteric-biased agonists have been developed, and intracellular-biased agonists have recently attracted broad interest. These agonists bind within the receptor intracellular cavity and preferentially tune the specific signalling pathway over other signalling pathways, without allosteric rearrangement of the receptor from the extracellular side[1–3]. However, only antagonist-bound structures are currently available[1,4–6], and there is no evidence to support that biased agonist binding occurs within the intracellular cavity. This limits the comprehension of intracellular-biased agonism and potential drug development. Here we report the cryogenic electron microscopy structure of a complex of $G_s$ and the human parathyroid hormone type 1 receptor (PTH1R) bound to a PTH1R agonist, PCO371. PCO371 binds within an intracellular pocket of PTH1R and directly interacts with $G_s$. The PCO371-binding mode rearranges the intracellular region towards the active conformation without extracellularly induced allosteric signal propagation. PCO371 stabilizes the significantly outward-bent conformation of transmembrane helix 6, which facilitates binding to G proteins rather than β-arrestins. Furthermore, PCO371 binds within the highly conserved intracellular pocket, activating 7 out of the 15 class B1 GPCRs. Our study identifies a new and conserved intracellular agonist-binding pocket and provides evidence of a biased signalling mechanism that targets the receptor–transducer interface.

GPCRs comprise the largest family of human proteins and are involved in nearly all physiological processes. Consequently, they are targeted by more than 30% of marketed drugs[7]. Agonists bind to an extracellular orthosteric-binding pocket of GPCRs, which induces conformational changes and stabilizing the active conformation of the transmembrane domain (TMD). The orthosteric pockets have developed highly diverse shapes and sequences, which enables responses to a range of extracellular stimuli[8]. In addition to orthosteric agonists, numerous allosteric modulators have been generated, and previous structural studies have identified various allosteric pockets[9]. Compared with orthosteric sites, allosteric sites tend to have a greater variety of amino acid residues; therefore, allosteric ligands provide subtype specificity for receptors. Although previous structural studies have revealed precise and specific recognition modes for orthosteric and allosteric ligands by individual receptors,

conserved agonist-binding pockets across distinct receptor subtypes have not yet been discovered[10].

Most agonists activate multiple signalling pathways, and some of these signals occasionally induce adverse pharmacological effects. Biased agonists, which preferentially activate a specific intracellular transducer, have the potential to maximize therapeutic impact while reducing adverse effects[10]. Current biased agonists commonly bind to the extracellular half of the TMD, whereas agonists that bind to the intracellular side, particularly at the receptor–transducer interface, may be preferable for precise modulation of biased signalling action[2]. Thus far, six structures of intracellular ligand-bound GPCRs have been reported[1,4–6,11,12]. However, most of these are antagonist-bound structures, and there is no structural evidence of biased agonists binding to an intracellular transducer pocket[1,4–6]. This lack of knowledge limits the ability to understand and fine-tune this intracellular-biased agonism.

[1]Department of Biological Sciences, Graduate School of Science, The University of Tokyo, Tokyo, Japan. [2]Graduate School of Pharmaceutical Sciences, Tohoku University, Sendai, Japan. [3]Research Division, Chugai Pharmaceutical, Shizuoka, Japan. [4]Laboratory of Organelle Pathophysiology, Department of Integrative Life Sciences, Graduate School of Life Sciences, Tohoku University, Sendai, Japan. [5]Department of Chemistry, Graduate School of Science, Chiba University, Chiba, Japan. [6]Present address: Preferred Networks, Tokyo, Japan. [7]These authors contributed equally: Kazuhiro Kobayashi, Kouki Kawakami. ✉e-mail: iaska@tohoku.ac.jp; t.murata@faculty.chiba-u.jp; nureki@bs.s.u-tokyo.ac.jp

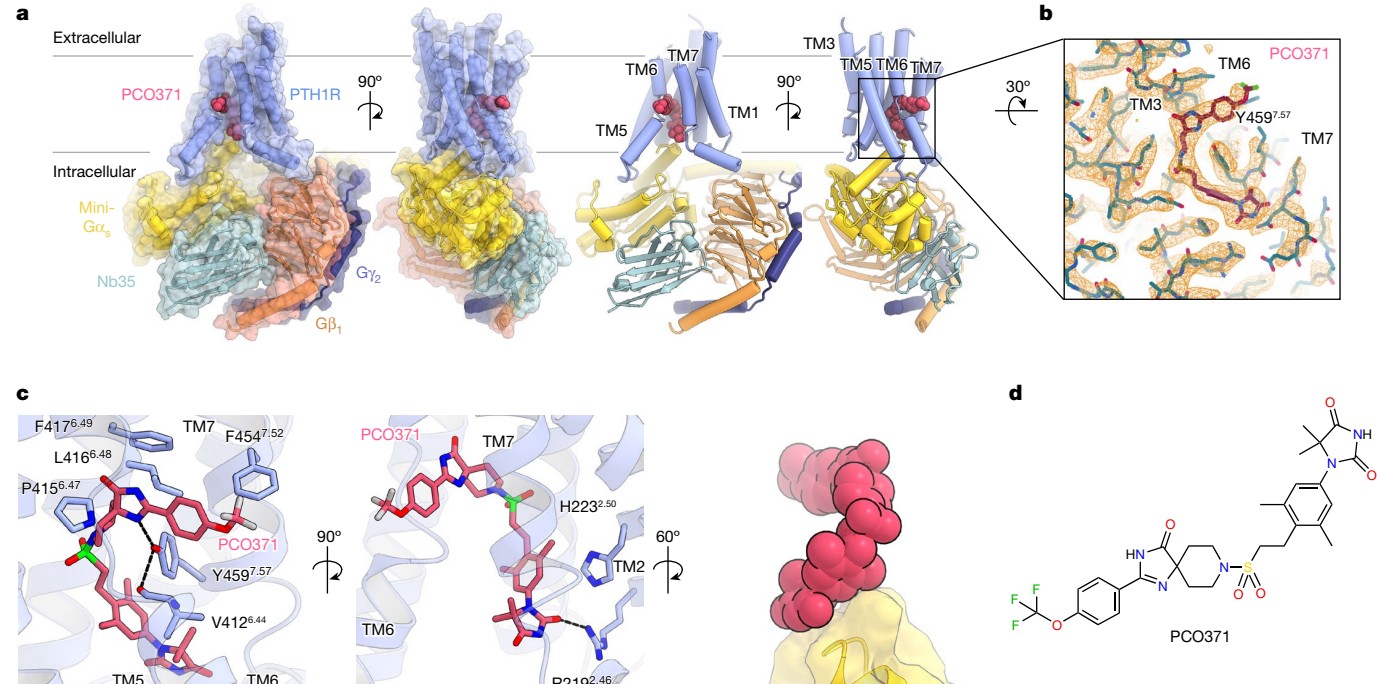

**Fig. 1 | Overall structure of PCO371–PTH1R–G$_s$. a**, Orthogonal views of the PCO371–PTH1R–G$_s$ complex, constructed from the cryo-EM potential map and coloured according to the subunit. Violet, PCO371-bound PTH1R; magenta, PCO371; yellow, mini-Gα$_s$ Ras-like domain; tomato, Gβ$_1$; navy, Gγ$_2$; powder blue, Nb35. **b**, Density map and constructed model of PCO371 near the PCO371-binding pocket. **c**, Close-up view of the PCO371-binding site. Numerical superscripts indicate relative positions in the receptor according to the Wootten class B1 GPCR numbering of the receptor TMD region[33]. The map is shown in the 2.085 e A$^{-3}$ counter level. **d**, Chemical structure of PCO371. PCO371 is composed of four chemical groups (shown from left to right): trifluoromethoxyphenyl, spiro-imidazolone, dimethylphenyl and DMH.

PTH1R is a class B1 GPCR and is a major regulator of mineral ion homeostasis and bone metabolism. PTH1R responds to parathyroid hormone (PTH) and parathyroid hormone-related peptide (PTHrP) ligands and activates G$_s$, G$_q$ and β-arrestins[13]. Natural and modified forms of these ligands induce substantial anabolic bone formation and are used for the clinical treatment of osteoporosis[14,15]. However, these ligands also induce catabolic bone resorption, which causes adverse effects. The balance between therapeutic and adverse effects depends on the differences in the duration of G$_s$-mediated cyclic AMP production[16]. Pulse G$_s$ activation at the plasma membrane induces transient cAMP production. Conversely, activated PTH1R is internalized by β-arrestins and induces sustained cAMP production at the early endosome, which leads to adverse effects. Thus, G-protein-biased agonists are useful in osteoporosis treatment; however, the mechanisms of G-protein-biased activation are largely unknown in PTH1R.

We have previously reported the structures of PTH and PTHrP–PTH1R–G$_s$ complexes, providing structural insights into why adverse effects occur[17]. Although this information is crucial for peptide-based drug development that targets PTH1R, peptides need to be administered by subcutaneous injection, and non-peptidic and orally administrated agonists are desirable to reduce the physical burden for patients. Recently, PCO371, a non-peptidic chemical PTH1R agonist, was shown to exhibit PTH-mimetic activity in vivo through oral administration[18]. However, there is no structural information for PCO371-bound PTH1R; thus, the binding site and activation mechanism of PCO371 remains to be elucidated.

## Non-canonical features of PCO371–PTH1R–G$_s$ complex

To elucidate how PCO371 binds to PTH1R and induces its conformational rearrangement, we determined the structure of the PCO371–PTH1R–G$_s$ signalling complex. PTH1R was expressed in HEK293 cells, solubilized in a solution of lauryl maltose neopentyl glycol (LMNG) with cholesteryl hemisuccinate (CHS) and purified in glyco-diosgenin (GDN) with the CHS solution in the presence of PCO371. We used a pH 9.0 solution during purification owing to the solubility requirements of PCO371 at high concentrations (Methods). PCO371 significantly increased G$_s$ activity under mildly basic conditions (pH 9.0), whereas PTH-induced G$_s$ activation was equivalent at both physiological pH (pH 7.4) and pH 9.0 (Extended Data Fig. 1). PCO371-bound PTH1R was mixed with the engineered mini-G$_s$ heterotrimer and nanobody 35 (Nb35) to form a Nb35-stabilized PCO371–PTH1R–mini-G$_s$ complex. After purification, this signalling complex was imaged using a Titan Krios G4 cryogenic transmission electron microscope with a K3 detector. Particle images were categorized by 2D and 3D classifications and then used to create a cryogenic electron microscopy (cryo-EM) density map at a global resolution of 2.9 Å (Fig. 1a, Extended Data Fig. 2 and Extended Data Table 1). This primary map enabled unambiguous assignment of the secondary structure and the side-chain orientations of the PCO371–PTH1R–G$_s$ complex, except for the extracellular domain of PTH1R (Fig. 1b and Extended Data Fig. 3). No clear density corresponding to the extracellular domain was observed, which was in contrast to the previously determined PTH–PTH1R–G$_s$ complex structure and several structures of small chemical agonist-bound glucagon-like peptide 1 receptor (GLP-1R)–G$_s$ complex (Extended Data Fig. 4a,b). This structural difference indicates that the extracellular domain is highly flexible and does not require a specific conformational state to allow G$_s$ engagement with PTH1R.

Notably, the PTH1R complex lacked any apparent ligand density in the orthosteric-binding pocket, the features of which have not been previously reported for most of the other agonist-bound class B1 GPCR structures[19] (Extended Data Fig. 4b). Instead, we observed a clear density corresponding to PCO371 in an intracellular-transducer-binding pocket formed by transmembrane helix 2 (TM2), TM3, TM6 and TM7,

which was spatially separate from the extracellular orthosteric pocket (Fig. 1a–c and Extended Data Fig. 4b). Most of the ligand–receptor contacts were mediated by van der Waals and hydrophobic interactions, consistent with the hydrophobic nature of PCO371 (Extended Data Table 2). Notably, $Tyr459^{7.57}$ (superscript numbers reflect the Wootten class B1 GPCR numbering) formed hydrogen bonds with the main-chain carbonyl group of $Val412^{6.44}$ and PCO371, which stabilizes the PCO371–PTH1R–$G_s$ complex (Fig. 1c). Additionally, the side chain of $Tyr459^{7.57}$ sequestered PCO371 from membrane lipids (Fig. 1a,b). The carbonyl group of dimethylhydantoin (DMH) on PCO371 formed a salt bridge with $Arg219^{2.46}$ (Fig. 1a,c,d). Notably, the DMH group of PCO371 (Fig. 1d) interacted with the carboxy-terminal hook of $G\alpha_s$, which tightly packed PCO371 into the receptor core (Fig. 1a,d). These interactions are consistent with the findings from our previous chemical structure–activity relationship study, which revealed that the absence of the PCO371 DMH group led to a significant decrease in potency[20,21]. Our structural observations, along with the previous chemical structure–activity relationship findings, suggest that PCO371 is an agonist that binds to $G_s$ and the intracellular cavity of the receptor.

## PCO371 tunes only the intracellular portion

To characterize the overall structure of the PCO371–PTH1R–$G_s$ complex, we compared this structure with those of PTH–PTH1R–$G_s$ and engineered PTH (ePTH)-bound PTH1R[17,22]. The PTH-bound PTH1R structure possessed the hallmark features of active class B1 GPCR structures. In brief, the extracellular portion of this structure exhibited the inwards movement of TM1, TM7 and the third extracellular loop, as well as the outwards movement of TM6, which induced the significant outwards movement of TM6 in the intracellular portion. Conversely, the ePTH-bound PTH1R structure, the only inactive-like PTH1R structure, exhibited inactive conformations in the TMD core and intracellular portion, although the extracellular portion was slightly shifted towards an active conformation owing to the binding of an engineered agonist (Fig. 2a,b).

Comparisons among the structures of the PCO371–PTH1R–$G_s$, PTH–PTH1R–$G_s$ and ePTH-bound PTH1R structures revealed the distinct conformation of the PCO371-induced active state. First, the extracellular portions of TM1, TM6 and TM7 in PCO371-bound PTH1R shifted by approximately 6 Å outwards, 6 Å inwards and 4 Å outwards, respectively, compared with the PTH-bound active PTH1R structure (Fig. 2a,b). These portions of TM1 and TM7 moved further outwards, and the portion of TM6 moved further inwards compared with ePTH-bound PTH1R. These movements are consistent with the previously characterized inactive–active conformational change in the class B1 GPCR structures[19,22–25] (Fig. 2b and Extended Fig. 5a), indicating that the extracellular portions of the TMD adopt the inactive conformation in the PCO371-bound PTH1R structure. In contrast to the extracellular portion, the intracellular portion of the TMD in PCO371-bound PTH1R exhibited the outwards movement of TM6, which is similar to the movement in the active PTH–PTH1R–$G_s$ structure (Fig. 2c). This distinct extracellular conformation caused TM6 to unwind and become moderately kinked (approximately 145°) in the PCO371-bound PTH1R structure, which is distinct from the sharply kinked TM6 (approximately 90°) in the PTH-bound PTH1R structure (Extended Data Fig. 5b). Superimposition of the PCO371-bound PTH1R and the PTH-bound PTH1R structures revealed that PCO371 spatially clashed with the sharply kinked TM6 conformation in PTH-bound PTH1R (Fig. 2d). Furthermore, the moderately kinked TM6 conformation of PCO371-bound PTH1R spatially clashed with the amino terminus of PTH in the PTH-bound PTH1R structure (Fig. 2d), consistent with previous ligand-competition experiments[18]. Superimposition of active class B1 GPCR structures revealed the mostly identical kink angles of TM6 (approximately 90° to 100°) in active class B1 GPCR structures bound to endogenous or chemical agonists (Fig. 2e, left, and Extended Data Fig. 5c, top and

middle). Thus, the moderate kink (approximately 145°) in TM6 caused by PCO371 binding is distinct from other active structures, and the angle is intermediate between the angles in the active (92°) and inactive-like (164°) PTH1R structures (Fig. 2e, centre and right, and Extended Data Fig. 5c, bottom). These results indicate that PCO371 directly binds to the intracellular transducer pocket and acts as a molecular wedge (Extended Data Fig. 5d). This binding position stabilizes the significant outwards movement of the intracellular portion of TM6 without signal propagation from the extracellular side, which is distinct from any of the previously analysed class B1 agonists and GPCR pairs.

## Activation mechanism of PCO371 compared with PTH

To gain insight into the mechanism of PCO371-induced PTH1R activation, we compared three key motifs among class B1 GPCRs: the PYQ ($Pro^{6.47}$–$Tyr^{6.53}$–$Gln^{7.49}$) active motif; the PxxG ($Pro^{6.47}$–x–x–$Gly^{6.50}$) switch; and the HETY ($His^{2.50}$–$Glu^{3.50}$–$Thr^{6.42}$–$Tyr^{7.59}$) inactive motif[17,26]. PTH bound to the orthosteric pocket and allosterically induced rearrangement to reconstruct the hydrogen-bonding network of the PYQ active motif (Extended Data Fig. 6a,b, left). These conformational changes facilitated the sharp kink in TM6 at $Gly418^{6.50}$ in the PxxG motif, relocating $Leu416^{6.48}$ and $Phe417^{6.49}$ (Extended Data Figs. 6c,d). The movement of $Leu416^{6.48}$ and $Phe417^{6.49}$ created a hydrophobic cluster and stabilized the outwards conformation of the intracellular half of TM6 (Extended Data Fig. 6d). Consequently, the relocated $Leu416^{6.48}$ pushed $Tyr459^{7.59}$, collapsing the HETY inactive motif and opening the intracellular cavity for G protein accommodation (Extended Data Fig. 6e).

The TMD of PCO371-bound PTH1R deformed the PYQ active motif and the PxxG switch of TM6 in the helical structure, in a similar manner to the ePTH-bound PTH1R structure (Figs. 2e and 3a–c and Extended Data Fig. 6f,g). In contrast to the inactive conformations of the PYQ and PxxG active motifs, trifluoromethoxyphenyl and the spiro-imidazolone groups of PCO371 (Fig. 1d) severely clashed with $Val412^{6.44}$–$Phe417^{6.49}$ and the HETY inactive motif in the ePTH-bound PTH1R structure, respectively (Extended Data Fig. 6h,i). To avoid this steric problem, our structure showed that TM6 in PCO371-bound PTH1R was unwound at $Pro415^{6.47}$, which indicated that this residue is necessary for the PCO371-specific activation of PTH1R. Consistent with these structural observations, a mutation in the PYQ motif ($Q451^{7.49}A$) and those in the PxxG switch ($L416^{6.48}A$, $F417^{6.49}A$ and $G418^{6.50}L$) selectively reduced PTH-induced cAMP accumulation but did not affect PCO371 activity (Fig. 3d and Extended Data Fig. 7). By contrast, the $P415^{6.47}L$ mutation selectively and completely abolished PCO371 activity but showed no significant effect on PTH activity (Fig. 3d and Extended Data Fig. 7). These results indicate that $Pro415^{6.47}$ is essential for the PCO371-mediated activation of PTH1R without the conformational rearrangement of the PYQ motif and the PxxG switch.

Generally, the TM6 conformation is crucial for the preferential binding of specific transducers and biased signalling. Previous studies have shown that the preferential binding of G proteins requires the significant outwards movement of TM6 and a large intracellular cavity, whereas for β-arrestins, it requires a slight outwards movement of TM6 and a small intracellular cavity[27–29]. PCO371 acts as a molecular wedge that stabilized the significantly outwards conformation of intracellular TM6 (Fig. 3e and Extended Data Figs. 5d and 8a,b), which indicated that PCO371 preferentially activates G proteins rather than β-arrestins. Consistent with our structural findings, the NanoBiT-based assays revealed that PCO371 binding elicited negligible recruitment signals for β-arrestin 1 and β-arrestin 2 (Fig. 3f). Subsequently, to thoroughly demonstrate that PCO371 is a G-protein-biased agonist, we observed the intracellular translocation of PTH1R–β-arrestin by confocal microscopy. We expressed PTH1R fused to a Flag-tag and β-arrestin 2 fused to mVenus in HEK293 cells. The receptor was labelled with an Alexa-647-fused Flag-M1 antibody, and their membrane and

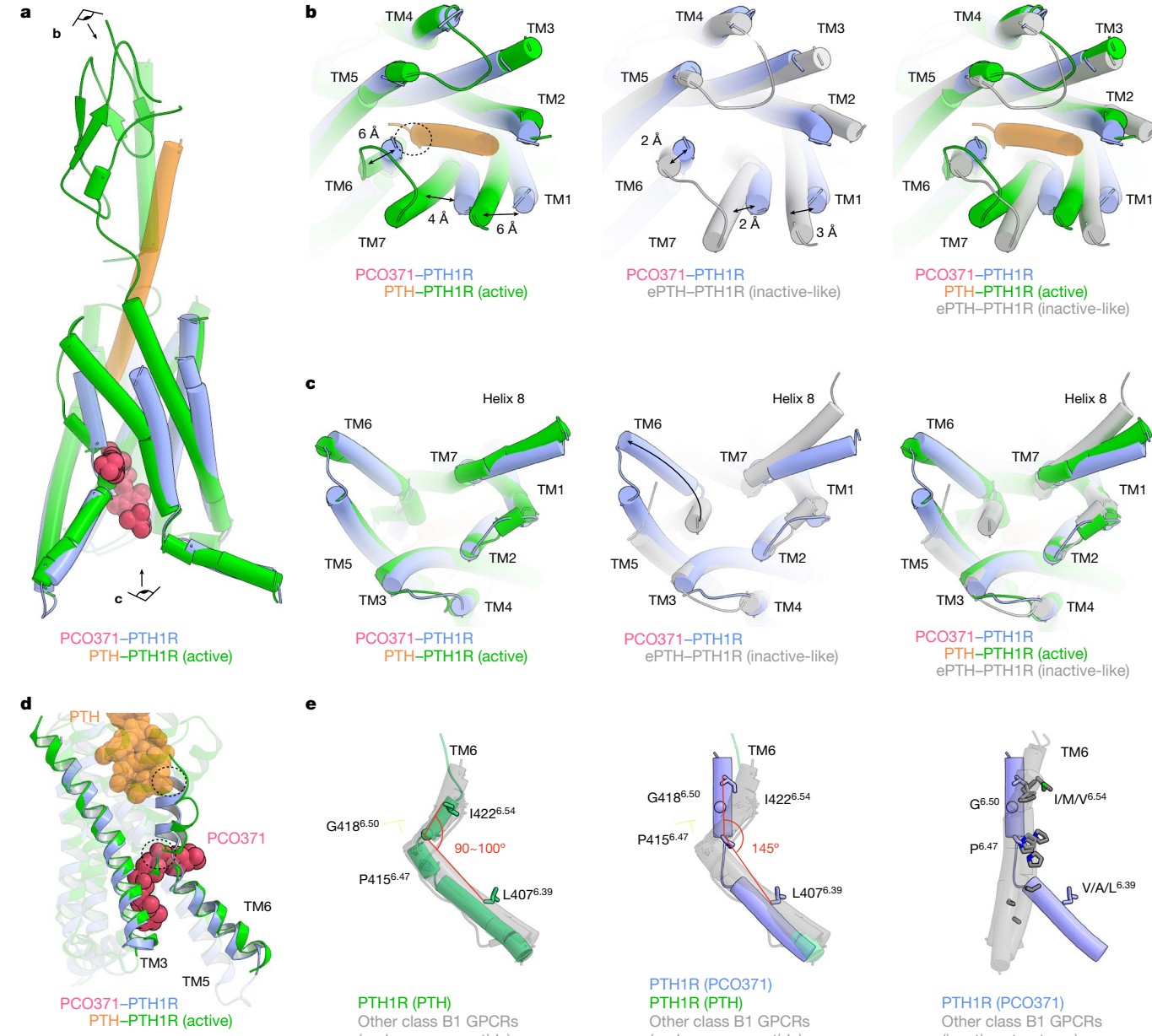

**Fig. 2 | Comparison of the PCO371-bound PTH1R structure with PTH-bound active PTH1R and ePTH-bound inactive-like PTH1R structures.**
**a**, Superimposition of PCO371–PTH1R–G$_s$ and PTH–PTH1R–G$_s$ (class 2, a representative active form) complexes, aligned on TM2–TM5. The eye and arrow symbols indicate angles of view in **b** and **c**. **b**,**c**, Extracellular (**b**) and intracellular (**c**) views of the superimposed TMDs of PCO371-bound active PTH1R, PTH-bound active PTH1R and ePTH-bound inactive PTH1R. **b**, Two-way arrows indicate distances of Cα atoms of Thr192$^{1.44}$ (TM1), Ile422$^{6.54}$ (TM6) and Met445$^{7.43}$ (TM7) residues between the structures of PCO371-bound PTH1R and PTH-bound active PTH1R, or between the structures of PCO371-bound and ePTH-bound inactive-like PTH1R. **c**, The one-way arrow indicates the typical

outwards movement of TM6 in PCO371-bound PTH1R and PTH-bound PTH1R. **d**, Allosteric competitive binding mechanism of PCO371 and PTH. PTH and PCO371 clash at the dashed circles with the other ligand-bound conformation. **e**, Superimposed structures of PTH-bound PTH1R, PCO371-bound PTH1R, inactive or inactive-like class B1 GPCRs (PTH1R, GCGR, GLP-1R and CRF1R), and all other structures of endogenous agonist-bound class B1 GPCR (PTH2R, SCTR, GHRHR, PAC1R, VIP1R, VIP2R, GCGR, GIPR, GLP-1R, GLP-2R, CALCR, CALRL, CRF1R and CRF2R). The angle is shown among the Cα atoms of Ile/Met/Val$^{6.54}$, Gly$^{6.50}$, and Val/Ala/Leu$^{6.39}$ residues in TM6. Note that the angle of PCO371-bound PTH1R is calculated among Leu$^{6.39}$, Pro$^{6.47}$ and Ile$^{6.54}$ owing to the kink at Pro$^{6.47}$.

subcellular localization were visualized by confocal microscopy. Intracellular co-localization of PTH1R and β-arrestin 2 occurred 30 min after stimulation with a high concentration (100 nM) and a low concentration (10 pM) of PTH (Fig. 3g and Extended Data Fig. 8c). By contrast, this intracellular co-localization was not observed following stimulation with a high concentration (100 μM) of PCO371, which is equivalent to 10 pM PTH stimulation in the cAMP assay (Fig. 3g and Extended Data Fig. 8c). Taken together, we conclude that PCO371 is a G-protein-biased agonist. Our study provides structural evidence that the intracellular

agonist PCO371 selectively activates G proteins and that a directly biased signalling mechanism can be induced without allosteric signal transduction.

## PCO371 is a potential seed for class B1 GPCRs

Our structure showed that PCO371 interacts with the intracellular pocket formed by the highly conserved residues of class B1 GPCRs (Arg219$^{2.46}$, His223$^{2.50}$, Leu226$^{2.53}$, Ile299$^{3.47}$, Glu302$^{3.50}$, Leu306$^{3.54}$,

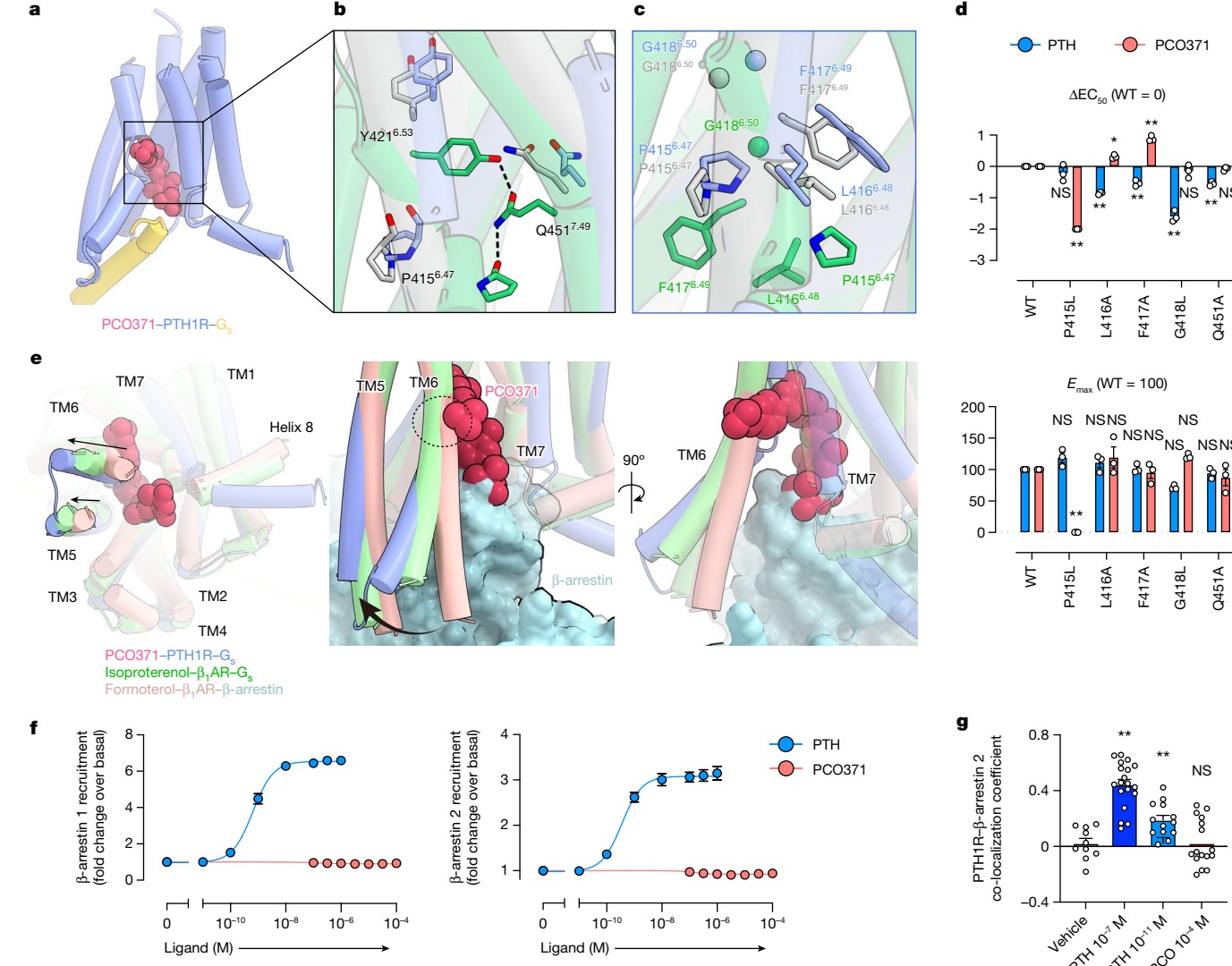

**Fig. 3 | Distinct activation mechanism of PTH1R induced by PCO371 and PTH. a**, TMD region of PCO371-bound PTH1R. **b,c**, Magnified view of the PYQ active motif and PxxG active switch. **d**, GloSensor cAMP responses in wild-type (WT) PTH1R and the PYQ motif or the PxxG switch mutants following PTH or PCO371 stimulation. The negative logarithmic half-maximal effective concentration (pEC₅₀) and the maximum response ($E_{max}$) values were calculated from the concentration–response curves (Extended Data Fig. 7a,c). *$P < 0.05$ and **$P < 0.01$, calculated using two-way analysis of variance (ANOVA) followed by Dunnett's test for multiple comparison analysis (with reference to the WT). NS, not significantly different between the groups. **e**, Superimposition of PCO371–PTH1R–$G_s$, isoproterenol–$\beta_1$ adrenaline receptor ($\beta_1$AR)–$G_s$ and formoterol–$\beta_1$AR–$\beta$-arrestin complexes, aligned on TM2–TM5. Black arrows indicate the hallmark conformational changes of TM5 and TM6 in $G_s$-bound and

$\beta$-arrestin-bound structures (left). PCO371 clashes with TM6 in the dashed circle (centre and right). **f**, Concentration–response curves of $\beta$-arrestin 1 and $\beta$-arrestin 2 recruitment to PTH1R following stimulation with PTH or PCO371. **g**, Co-localization of PTH1R–$\beta$-arrestin 2 in response to PCO371 and different concentrations of PTH. Quantification of co-localization of PTH1R and $\beta$-arrestin 2 after stimulation with vehicle, 100 nM PTH, 10 pM PTH or 100 µM PCO371. The co-localization index for individual cells in each stimulation condition was calculated using Fiji (ImageJ). Symbols and error bars represent the means and s.e.m., respectively, of 10–19 cells. *$P < 0.05$ and **$P < 0.01$, calculated using one-way ANOVA followed by Dunnett's test for multiple comparison analysis with reference to the vehicle stimulation. Data are from three independent experiments (**d,f**).

Val412⁶·⁴⁴, Leu413⁶·⁴⁵, Met414⁶·⁴⁶, Pro415⁶·⁴⁷, Leu416⁶·⁴⁸, Phe417⁶·⁴⁹, Gly418⁶·⁵⁰, Phe454⁷·⁵², Tyr459⁷·⁵⁷ and Asn463⁸·⁴⁷) (Fig. 4a,b). To evaluate potential applications involving this pocket, we measured the levels of PCO371-induced cAMP production in all class B1 GPCRs using a cAMP accumulation assay. Remarkably, PCO371 activated 7 out of 15 class B1 GPCRs (Fig. 4c and Extended Data Fig. 8d). We compared GPCR sequence similarity by a phylogenetic tree analysis[30] (calculated using GPCRdb (https://gpcrdb.org/) and found that PCO371 activates receptors assigned to the same group, the PTH1R clade, with the exceptions of PTH2R and the glucagon receptor (GCGR) (Fig. 4d).

To determine why PCO371 failed to activate PTH2R, we investigated the non-conserved leucine residue at the position 6.47. Given that class

B1 GPCRs broadly adopt the proline residue at position 6.47, which has a crucial role in PCO371 recognition of PTH1R (Fig. 3d and Extended Data Fig. 6f), we examined whether the lack of response to PCO371 is caused by the different amino acid at the 6.47 position in PTH2R. Consistent with our hypothesis, the L370⁶·⁴⁷P PTH2R mutant generated a cAMP accumulation in response to PCO371 (Fig. 4e and Extended Data Fig. 8e). Considering that PTH activated both wild-type and P415⁶·⁴⁷L PTH1R, whereas PCO371 did not activate P415⁶·⁴⁷L PTH1R (Fig. 3d), we concluded that Pro⁶·⁴⁷ is an essential determinant for PCO371-induced receptor activation.

Our work revealed that PCO371 is a versatile agonist for seven class B1 GPCRs (Fig. 4c,d), which suggests that this orally available phase 1

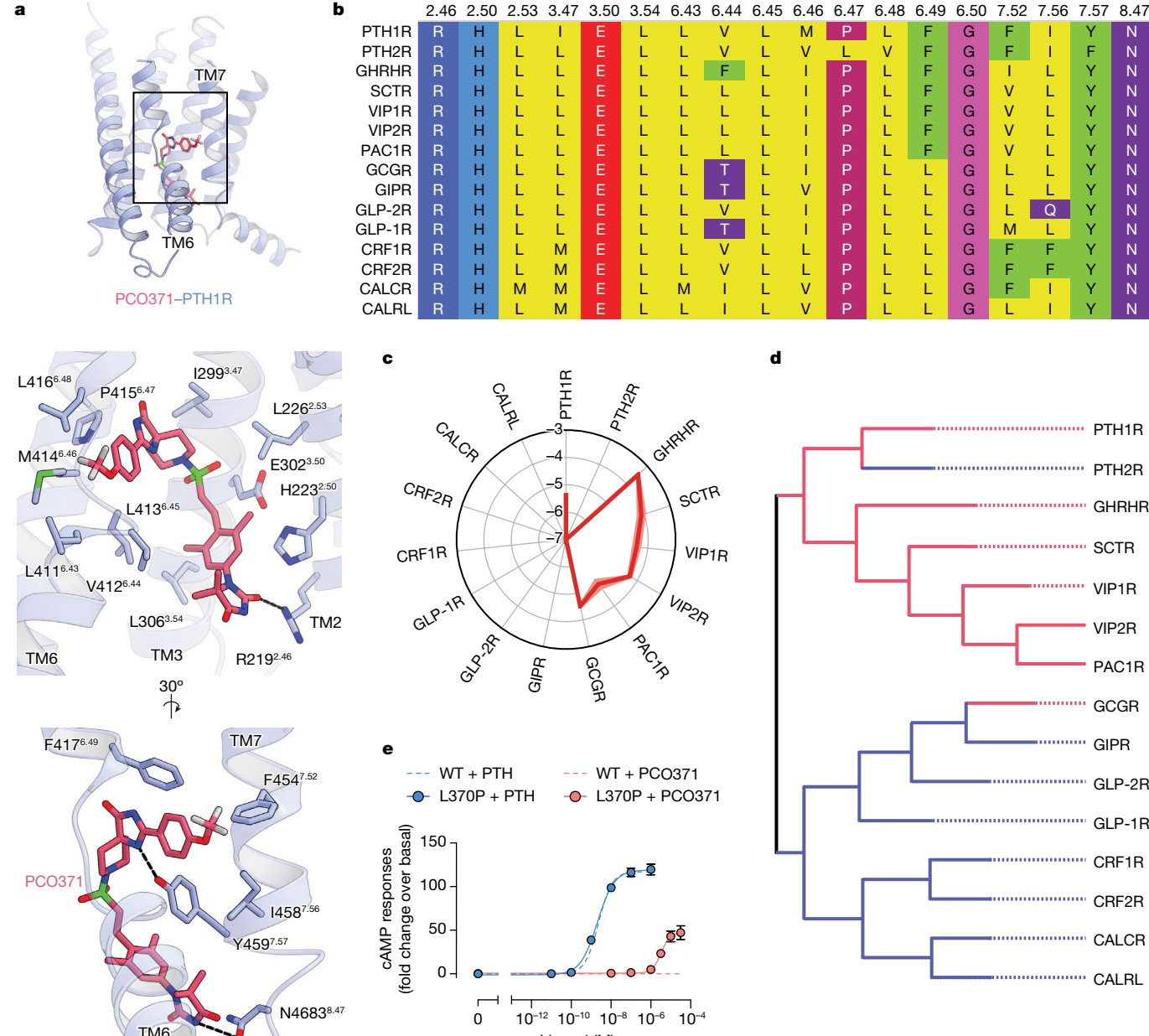

**Fig. 4 | PCO371 activates nearly half of class B1 GPCRs. a**, TMD region of PCO371-bound PTH1R (top) and magnified view of the PCO371-binding region (middle and bottom). **b**, Amino acid sequence alignment of human class B1 GPCRs. Described residues interact with PCO371. **c**, PCO371-induced GloSensor cAMP responses among class B1 GPCRs. Values in the radar chart indicate the logarithmic values of relative intrinsic activity ($\Delta \log RIA_{PCO\text{-}peptide}$), which is defined as the $E_{max}/EC_{50}$ value ($RIA_{PCO}$) in each receptor normalized by the $E_{max}/EC_{50}$ value following stimulation by its endogenous peptide agonists ($RIA_{peptide}$). Lines and shaded regions represent the means and s.e.m., respectively, of three independent experiments with each performed in duplicate. Note that in eight

PCO371-insensitive GPCRs (PTH2R, GIPR, GLP-2R, GLP-1R, CRF1R, CRF2R, CALCR and CALRL), the $RIA_{PCO}$ values could not be calculated. Therefore the $\Delta \log RIA_{PCO\text{-}peptide}$ values are denoted as less than $-7$. **d**, Phylogenetic tree of class B1 GPCRs. PCO371-sensitive and PCO371-insensitive receptors are indicated with red and blue lines, respectively. PCO371 activates members of the PTH1R clade, except PTH2R and GCGR. **e**, Concentration–response curves of GloSensor cAMP responses of WT and L370[6.47]P mutant (L370P) PTH2R following stimulation with PTH or PCO371. Symbols and error bars represent the means and s.e.m., respectively, of three independent experiments performed in duplicate.

drug candidate may be a promising drug seed to target these receptors. However, questions remain regarding the receptor selectivity of PCO371 between the PAC1R clade and GLP-1R clades (Fig. 4d). Based on structural comparisons with compound 2-bound GLP-1R, we speculate that the concomitant movement of the extracellular and intracellular halves of TM6 is a key feature for specific activation. Compound 2 is an intracellular agonist that binds to the outside of intracellular TM6 (Extended Data Fig. 9a,b, left). The compound-2-bound GLP-1R structure exhibited the typical sharp kink in TM6, similar to the orthosteric

peptide-bound active structures[11] (Fig. 2e and Extended Data Fig. 9b,c). The extracellular conformation of GLP-1R is presumably rearranged in conjunction with the conformational change on the intracellular side. Although this inside-out conformational change in GLP-1R is a common feature observed in other GPCRs, such as class A GPCRs, an extracellular conformational change was not observed in PCO371-bound PTH1R. Thus, we proposed that PCO371 only activates a group of GPCRs that do not require the inside-out change of TM6 for receptor activation because PCO371 is unable to bind receptors that adopt the typically

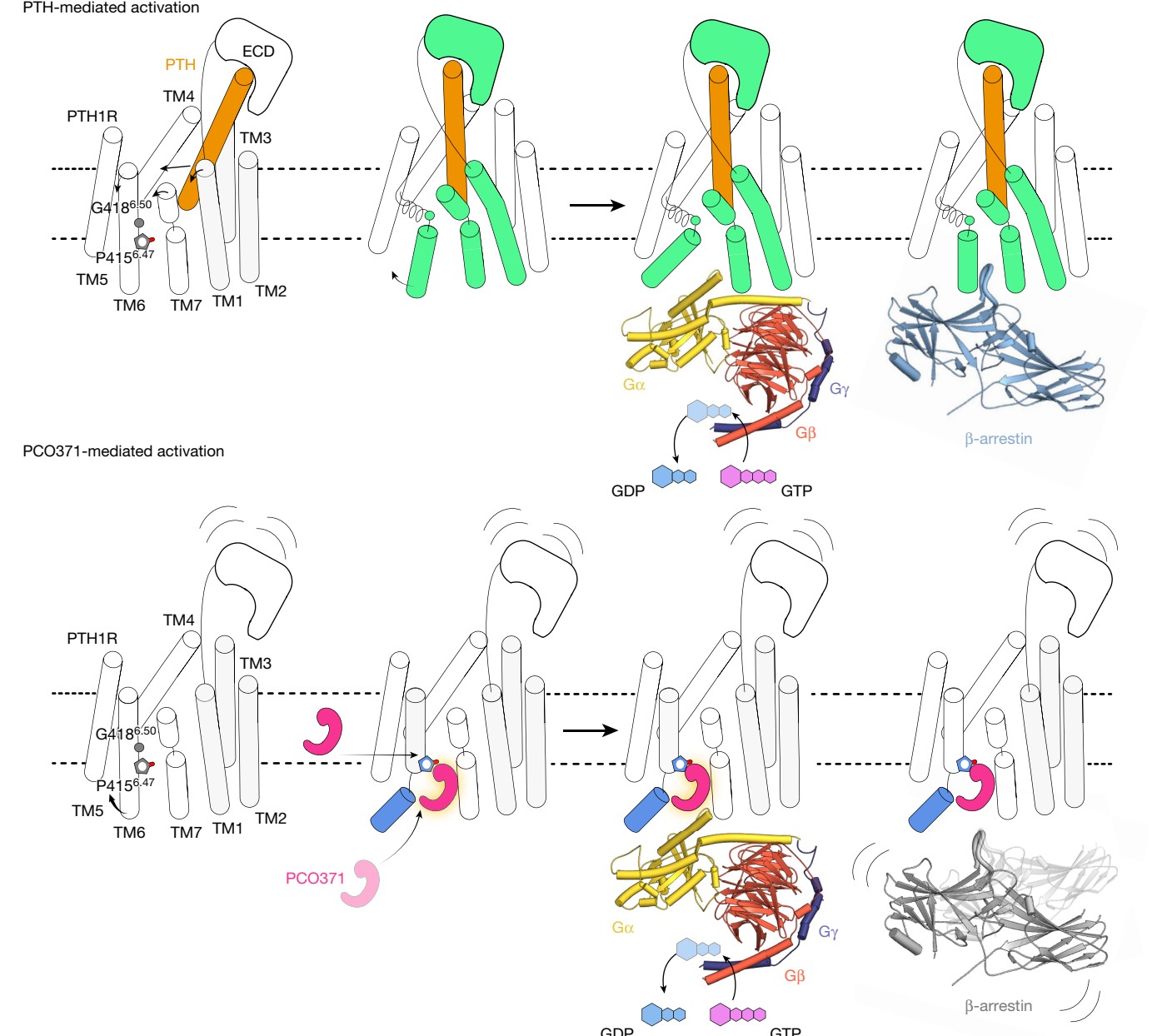

**Fig. 5 | Proposed mechanisms of PCO371-induced activation and signalling compared with known mechanisms of PTH-induced activation and signalling.** The distinct activation and functional selectivity mechanism of PTH1R. Top, PTH induces the rearrangement of the extracellular portions of TM1, TM6 and TM7, which causes the outwards movement of the intracellular portion of TM6 and the formation of a kink at Gly418[6.50]. PTH-bound PTH1R can adopt preferential conformations for G proteins and β-arrestins, respectively. Bottom, PCO371 directly moves the intracellular portion of TM6 outwards and causes the formation of a moderate kink in TM6 at Pro415[6.47] without requiring extracellular rearrangement. PCO371-bound PTH1R solely adopts the preferential conformation for G proteins by stabilizing the outwards conformation of the intracellular portion of TM6. ECD, extracellular domain.

activated extracellular conformation (Fig. 2d). To prove this hypothesis, we designed three chimeric receptors—PTH1R (TM6 replaced by that of GLP-1R), PTH1R (TM6 replaced by that of PAC1R) and GCGR (TM6 replaced by that of GLP-1R)—and measured cAMP production induced by PCO371. Notably, these receptors showed only minor differences in their endogenous ligand responses, which suggested that these chimeric receptors retain their receptor activities (Extended Data Fig. 9d). Moreover, PTH1R (TM6-GLP-1R) completely abolished cAMP production by PCO371, whereas PTH1R (TM6-PAC1R) showed distinguishable cAMP production by PCO371 (Extended Data Fig. 9d). In line with our notion, GCGR (TM6-GLP-1R) also mostly lacked cAMP production by PCO371 (Extended Data Fig. 9d). These results strongly support our proposal that PCO371 activates receptors that can independently mobilize the intracellular half of TM6 from the extracellular half.

## Discussion

In this study, we identified a new intracellular agonistic pocket of PTH1R and an atypical activation process in which PCO371 acts as a molecular wedge. Previously reported class B1 GPCR structures have revealed the common mechanisms of allosteric signal transduction by peptides and small chemical agonists that bind to the orthosteric site, although the binding modes differ between chemical agonists and peptide agonists (Extended Data Figs. 4 and 5). In contrast to these previously reported

ligands, PCO371 directly binds to the intracellular transducer pocket of PTH1R and activates it without requiring allosteric signal transduction from the extracellular side (Fig. 5). These results suggest that PCO371 successfully traverses the cell membrane and directly accesses the intracellular side or intramembrane region like a membrane lipid (Fig. 5). Instead of inducing the typical sharp kink in TM6 at Gly418[6.50], PCO371 unwinds TM6 at Pro415[6.47], which is necessary for its activity (Figs. 3e and 5 and Extended Data Fig. 6b,f). Our study revealed a distinct mechanism for GPCR activation, which expands the potential for drug development strategies.

Intracellular ligand pockets have received considerable interest in GPCR pharmacology because of their potential to selectively interact with and activate specific signal transducers. Thus far, the structures of multiple antagonists, positive allosteric modulators and an ago-allosteric modulator in complex with their corresponding GPCRs have been reported[1,4–6,11,12] (Extended Fig. 9a). Although these ligands bind with the intracellular region of each GPCR, there is no structure that accommodates an agonist in the intracellular transducer pocket, which limits structural insights concerning biased signalling from the intracellular side. Our structure provides a detailed view of an agonist bound in the intracellular cavity and directly interacting with a G protein. Given that G proteins require an open cavity conformation, whereas β-arrestins prefer a closed conformation[27–29], the size of this cavity is considered to be pivotal for biased agonism (Fig. 3g–i and Extended Data Fig. 8a,b). G-protein-biased agonism is beneficial for drug discovery with the class B1 GPCRs because class B1 GPCR-mediated β-arrestin activation generally induces prolonged signalling through the early endosome, which leads to adverse effects[31]. PCO371 directly interacts with the C-terminal hook of Gα$_s$, which exhibits divergent sequences among Gα subunits. Thus, modifications of PCO371 may gain further selective functionality among G protein subtypes (Extended Data Fig. 9e,f).

To further analyse PCO371 sensitivity in class B1 GPCRs, we superimposed and carefully inspected previously reported endogenous ligand-bound active class B1 GPCRs. We noted an alternative conformation of Asn[5.50] between GCGR and gastric inhibitory polypeptide receptor (GIPR), which is genetically similar but with different sensitivity for PCO371 (Extended Data Fig. 10a,b). The superimposed active class B1 GPCR structures showed that the PCO371-insensitive receptors adopted the outward Asn[5.50] conformation towards TM3, whereas the PCO371-sensitive receptors adopted the inwards conformation towards the centre of the receptor (Fig. 4d and Extended Data Fig. 10c). We also performed three independent molecular dynamics simulations, starting from our PCO371-bound PTH1R cryo-EM structure, and analysed the functionality of Asn[5.50] for PCO371 recognition. These simulations showed that the TMD of the receptor maintained an active conformation similar to the cryo-EM structure, and PCO371 was stably bound to the receptor (Extended Data Figs. 10d, top two panels). The Asn[5.50] conformation was stable in its position in our PCO371-bound PTH1R cryo-EM structure, which is distinct from that of the active GLP-1-bound GLP-1R structure (Protein Data Bank (PDB) identifier: 6X18) (Extended Data Fig. 10d, second from the bottom, and 10e). Notably, in the second run, the receptor showed an intermediate state, and PCO371 created a new and stable interaction with Asn374[5.50] (Extended Data Fig. 10d, bottom, 10f,g). Consistent with the simulation, the N374[5.50]A PTH1R mutant selectively reduced PCO371-induced receptor activation, whereas this mutant had no effect on PTH-induced receptor activation (Extended Data Fig. 10h). These results suggest that the side chain orientation of Asn374[5.50] is crucial for PCO371 recognition and that Asn374[5.50] can respond to PCO371 when its side chain is oriented towards the centre of the receptor (Extended Data Fig. 10a–c). Combined with our molecular dynamics simulations and functional study, these results suggest that Asn[5.50] is one of the determinants for PCO371 recognition and sensitivity in class B1 GPCRs. In the future, the structure of PCO371-bound GCGR may provide a comprehensive understanding of the receptor selectivity of PCO371.

In drug development, receptor specificity is crucial for therapeutic efficacy and safety. For GPCRs, the TMD core shares high sequence similarity, whereas the extracellular and intracellular loops have diverse sequences. These differences indicate that agonists gain receptor selectivity by extending the ligand towards the outside of the receptor[32]. Our structure revealed that the intracellular cavity of PCO371-bound PTH1R is tightly sealed by G$_s$, and PCO371 cannot interact with the intracellular loops (Extended Data Fig. 11a,b). Instead, the ligand-binding pocket of PCO371 is laterally open between TM6 and TM7, which is accessible to helix 8 (Extended Data Fig. 11b, left). Moreover, our structure visualized an intracellular pocket consisting of TM1, TM7 and helix 8, facing towards the membrane region. Given that helix 8 possesses sequence diversity, a binder molecule bound to this pocket may provide receptor selectivity, leading to the development of a bitopic ligand (Extended Data Fig. 11c,d). Moreover, PTH1R possesses a non-conserved cysteine residue at position 7.60 on the intracellular loop, connecting TM7 and helix 8 (Extended Data Fig. 11c, right). Note that only PTH1R, PTH2R and CALCR have the cysteine residue at position 7.60, and PCO371 does not activate CALCR and PTH2R. Thus, the addition of a covalent functional group to PCO371 may provide PCO371 receptor selectivity towards PTH1R. Cβ of Cys[7.60] is located 6.4 Å away from O6 of PCO371 (Extended Data Fig. 11c, right), and three or four carbon residues would be sufficient to connect PCO371 and Cys[7.60]. In addition, previously reported covalent agents (compound 2)[11] for GLP-1R may also be useful in the generation of bitopic ligands. Compound 2 acts as a weak covalent bond agent and selectively binds to Cys347[6.36]. By optimizing the binding of compound 2 to the cavity formed by TM6, TM7 and Gα$_s$ of PTH1R, a modified compound 2–PCO371 may become a PTH1R-selective agonist.

In summary, the distinct intracellular transducer pocket identified in this study may have broad potential in the development of biased chemical compounds for agonists, antagonists and allosteric modulators. Our findings will broaden the overall understanding of the mechanisms of GPCR activation and provide new strategies for the design and development of GPCR-targeted therapeutics.

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

## Methods

### Expression and purification of human PTH1R

The plasmid encoding human *PTH1R* (GenBank identifier: U17418.1; residues 27–491) was constructed and purified as previously reported[17]. The construct was expressed in HEK293 GnTI (N-acetylglucosaminyltransferase I-negative) cells (American Type Culture Collection, CRL-3022) using the BacMam system (Thermo Fisher Scientific), and the cells were grown and maintained in FreeStyle 293 medium (Gibco) at 37 °C, with 8% $CO_2$ under humidified conditions. Note that the cells were infected by the baculovirus generated in Sf9 cells (Life Technologies), supplemented with 10 mM sodium butyrate to boost protein expression after 18 h and cultivated in suspension at 30 °C for another 48 h. The cultured cells were collected by centrifugation (5,000$g$, 10 min, 4 °C) and disrupted by sonication in a hypotonic lysis buffer (20 mM Tris-HCl, pH 7.5, 150 mM NaCl and 20% glycerol). Cell debris was removed by centrifugation (10,000$g$, 10 min, 4 °C). The membrane fraction was collected by ultracentrifugation at 180,000$g$ for 1 h, and solubilized in a solubilization buffer (20 mM Tris-HCl, pH 9.0, 200 mM NaCl, 1% LMNG, 0.1% CHS, 20% glycerol and 100 μM PCO371) for 2 h at 4 °C. The solubilized receptors were separated from the insoluble material by ultracentrifugation at 180,000$g$ for 20 min and incubated with Ni-NTA resin (Qiagen) for 30 min. Detergent micelles were replaced by washing with 20 column volumes of wash buffer (20 mM Tris-HCl, pH 9.0, 500 mM NaCl, 0.03% GDN, 10% glycerol, 100 μM PCO371 and 30 mM imidazole). The receptor was eluted in elution buffer (20 mM Tris-HCl, pH 9.0, 500 mM NaCl, 0.01% GDN, 10% glycerol, 100 μM PCO371 and 300 mM imidazole). The eluate was treated with TEV protease to cleave the GFP-His$_{10}$ tag and dialysed against dialysis buffer (20 mM Tris-HCl, pH 9.0, 500 mM NaCl, 100 μM PCO371 and 10% glycerol). The cleaved GFP-His$_{10}$ tags and the TEV proteases were removed with Ni-NTA resin. The receptor was purified by size-exclusion chromatography on a Superdex 200 10/300 Increase column, equilibrated in SEC buffer (20 mM Tris-HCl, pH 9.0, 150 mM NaCl, 0.01% GDN and 100 μM PCO371). The peak fractions were collected and concentrated to about 5 mg ml$^{-1}$.

### Expression and purification of the mini-G$_s$ heterotrimer

The plasmid encoding mini-G$_s$ was constructed and purified as previously reported[34]. Mini-G$_s$ was expressed in *Escherichia* coli (BL21) cells. The cells were cultured in LB medium supplemented with 1 mM isopropyl-β-D-thiogalactopyranoside (IPTG) at 25 °C. After 20 h, the cells were disrupted by ultrasonication in hypotonic buffer (20 mM Tris-HCl, pH 7.5, 150 mM NaCl, 2 mM $MgCl_2$ and 1 μM GDP), and the mini-G$_s$ protein was purified by Ni-NTA affinity chromatography and then subjected to size-exclusion chromatography on a HiLoad Superdex75 16/600 column.

His$_6$-tag-fused rat Gβ1 and bovine Gγ2 were also constructed, expressed and purified as previously reported[17]. Cell cultures were grown to a cell density of $4 \times 10^6$ cells per ml in Sf900 II medium (Gibco) for 60 h at 27 °C. The cells were collected by centrifugation and lysed in hypotonic buffer. The Gβ1–Gγ2 heterodimer was purified by Ni-NTA affinity chromatography and then subjected to size-exclusion chromatography on a HiLoad Superdex75 16/600 column.

The purified mini-G$_s$ and Gβ1–Gγ2 were mixed and incubated overnight on ice. The mixture was concentrated and loaded onto a Superdex 75 10/300 Increase size-exclusion column and equilibrated in buffer (20 mM Tris-HCl, pH 7.5, 150 mM NaCl and 1 μM GDP). Peak fractions containing the mini-G$_s$ heterotrimer were pooled and concentrated to 5 mg ml$^{-1}$.

### Expression and purification of Nb35

The plasmid encoding Nb35 was prepared as previously reported[17,35]. The protein was expressed in the periplasm of *E. coli* C41 (Rosetta) cells cultured in LB medium supplemented with 1 mM IPTG for 20 h at 25 °C.

After 20 h, the cells were collected and disrupted by ultrasonication in hypotonic buffer (20 mM Tris-HCl, pH 7.5, 150 mM NaCl and 2 mM $MgCl_2$), and the Nb35 protein was purified by Ni-NTA affinity chromatography and then subjected to size-exclusion chromatography on a HiLoad Superdex75 16/600 column. Peak fractions were pooled and concentrated to 3 mg ml$^{-1}$.

### Formation and purification of the PCO371–PTH1R–mini-G$_s$β$_1$γ$_2$–Nb35 complex

PCO371 was synthesized at Chugai Pharmaceutical, and its purity and stability was confirmed by liquid chromatography–mass spectrometry. The purified PCO371-bound PTH1R proteins were mixed with a 1.2-fold molar excess of mini-Gsβ1γ2 and a 1.5-fold molar excess of Nb35, and the mixture was incubated on ice for overnight in pH 9.0. The sample was purified using Ni-NTA affinity resin and loaded onto a Superdex 200 10/300 Increase size-exclusion column, equilibrated in buffer (20 mM Tris-HCl, pH 9.0, 150 mM NaCl, 0.01% GDN and 100 μM PCO371) to separate the complex from contaminants. Peak fractions of the PCO371–PTH1R–mini-G$_s$ heterotrimer–Nb35 complex were pooled and concentrated to 7 mg ml$^{-1}$.

### EM data collection and processing

The purified complex was applied onto a freshly glow-discharged Quantifoil holey carbon grid (R1.2/1.3, Au, 300 mesh) using a Vitrobot Mark IV at 4 °C in 100% humidity. The prepared grids were transferred to a Titan Krios G4 microscope (Thermo Fisher Scientific), operated at an accelerating voltage of 300 kV with a Gatan Quantum-LS Energy Filter (GIF) and a Gatan K3 Summit direct electron detector in nanoprobe EFTEM mode. Images were collected at a nominal magnification of 105 K, corresponding to a calibrated pixel size of 0.83 Å per pixel (The University of Tokyo, Japan). The dataset was acquired using the Serial EM (v.3.7.4) software, with a defocus range of −0.8 to −1.6 μm. Each image was dose-fractionated to 72 frames at a dose rate of 7.5 e$^-$ pixel$^{-1}$ s$^{-1}$ to accumulate a total dose of 54.578 e$^-$ Å$^{-2}$. The 6,333 dose-fractionated movies were subjected to beam-induced motion correction using RELION-3 (ref. 36), and the contrast transfer function and the defocus parameters were estimated using CTFFIND4 (ref. 37). The 4,880,293 particles were initially picked from the 6,333 micrographs using the Laplacian-of-Gaussian picking function in RELION-3 and extracted in 3.63 Å per pixel. These particles were subjected to several rounds of 2D and 3D classifications. The best class contained 109,422 particles, which were then re-extracted with a pixel size of 1.11 Å per pixel and subjected to 3D refinement. The homogenous subset was subjected to per-particle defocus refinement, beam-tilt refinement, Bayesian polishing and 3D refinement. The final 3D refinement and post-processing produced maps with global resolution at 2.9 Å according to the Fourier shell correlation = 0.143 criterion. The processing strategy is described in Extended Data Fig. 2.

### Model building and refinement

The initial template for the PTH1R–G$_s$–Nb35 complex was derived from the PTH–PTH1R–G$_s$–Nb35 structure (PDB identifier: 7VVL), followed by extensive remodelling using COOT-0.9.3 EL[38]. The models of PCO371–PTH1R–G$_s$ were manually readjusted using COOT, refined using phenix.real_space_refine (v.1.14-3260)[39] REFMAC5[40] and Servalcat[41] against the working maps, and validated in MolProbity[42].

### Molecular dynamics simulation

The system included the PTH1R, PCO371, 1-phosphoryl-2-oleoylphosphatidylcholine (POPC), TIP3P water and 150 mM NaCl. The initial model of PTH1R containing amino acids 27–491 was created using MODELLER (v.10.1)[43], with the cryo-EM structure of PTH1R in complex with PCO371 as the template. The missing hydrogen atoms were built with the program VMD (v.1.9.3)[44]. The protein was embedded into the POPC membrane using the MemProtMD pipeline[45]. The net charge of

the simulation system was neutralized through the addition of 150 mM NaCl. The simulation system was $96 \times 96 \times 180$ Å$^3$ and contained 123,567 atoms. The molecular topologies and parameters from the Charmm36 force field[46] were used for the protein, lipid and water molecules. The molecular topology and parameters for PCO371 were prepared using the CHARMM-GUI ligand reader and modeller[47,48].

Molecular dynamics simulations were performed with the program NAMD (v.2.13). The simulation systems were energy minimized for 1,000 steps with fixed positions of the non-hydrogen atoms. After minimization, another 1,000 steps of energy minimization were performed with 10 kcal mol$^{-1}$ restraints for the non-hydrogen atoms, except for the lipid molecules within 5.0 Å of the proteins. Next, equilibrations were performed for 0.1 ns under NVT conditions, with 10 kcal mol$^{-1}$ Å$^{-2}$ restraints for the heavy atoms of the proteins. Finally, equilibration was performed for 2.0 ns under NPT conditions, with the 1.0 kcal mol$^{-1}$ Å$^{-2}$ restraints for all Cα atoms of the proteins. The production runs were performed for 200 ns without restraints while maintaining a constant temperature at 310 K using Langevin dynamics and constant pressure at 1 atm using a Nosé–Hoover Langevin piston[49,50]. The long-range electrostatic interactions were calculated using the particle mesh Ewald method[50]. The simulations were independently performed three times. The simulation results were analysed and visualized using mdtraj (v.1.9.8)[51], seaborn (https://zenodo.org/record/54844) and CUEMOL (v.2.2.3.443) (http://www.cuemol.org).

### Glo-sensor cAMP accumulation assay

PTH1R-induced $G_s$ activation was measured using the GloSensor cAMP accumulation assay. We first constructed a plasmid wherein the human full-length *PTH1R* gene was N-terminally fused to the Flag epitope tag with the preceding haemagglutinin-derived signal sequence. HEK293A cells were seeded in a 6-cm culture dish at a concentration of $2 \times 10^5$ cells per ml (4 ml per well in DMEM (Nissui Pharmaceutical) supplemented with 10% FBS (Sigma, F7524, lot 0001641439) and penicillin–streptomycin–glutamine (complete DMEM)) 1 day before transfection. The transfection solution was prepared by combining 10 µl (per well hereafter) of 1 mg ml$^{-1}$ polyethylenimine MAX (Polysciences) solution and a plasmid mixture consisting of 1,000 ng Glo-22F cAMP biosensor (human codon-optimized and gene-synthesized)-encoding pCAGGS plasmids and 400 ng Flag–PTH1R plasmids in 400 µl of Opti-MEM (ThermoFisher Scientific). After incubation for 24 h, transfected cells were collected using 0.53 mM EDTA-containing Dulbecco's PBS (D-PBS), centrifuged and suspended in 2 ml of HBSS containing 0.01% BSA (fatty-acid-free grade; SERVA) and 5 mM HEPES (pH 7.4) (assay buffer). The cell suspension was dispensed into a white 96-well plate at a volume of 40 µl per well and loaded with 10 µl of 10 mM D-luciferin potassium solution (FujiFilm Wako Pure Chemical) diluted in assay buffer. After 2 h of incubation at room temperature, the plate was measured for baseline luminescence (SpectraMax L equipped with 2PMT, Molecular Devices; SoftMax Pro (v.7.03), Molecular Devices) and 20 µl of 6× ligand diluted in the assay buffer or the assay buffer alone (vehicle) was manually added. The plate was read for 20 min with an interval of 30 s at room temperature. The luminescence counts over 8–10 min after ligand addition were averaged and normalized to the initial counts, and the fold changes in the signals over the vehicle treatment were further normalized to forskolin (10 µM) and plotted for the cAMP accumulation response. Using the Prism 9 software (GraphPad), the response were fitted to all data using the nonlinear regression. The variable slope (four parameter) in the Prism 9 tool with a constraint of the Hill slope of absolute value less than 2 was used. $pEC_{50}$ and $E_{max}$ values were obtained from the nonlinear regression curve of the averaged data. For multiple comparison analysis, one-way or two-way ANOVA and followed by Dunnett's test was used.

### NanoBiT β-arrestin recruitment assay

β-arrestin recruitment to PTH1R was measured using the NanoBiT β-arrestin-recruitment assay[52] with minor modifications. In brief,

plasmid transfection was performed in a 6-cm culture dish with a mixture of 200 ng N-terminal large BiT-fused β-arrestin 1 (Lg-β-arrestin 1) or Lg-β-arrestin 2 and 1,000 ng C-terminal small BiT-fused PTH1R (PTH1R-Sm) plasmids. After 24 h of incubation, the transfected cells were collected using 0.53 mM EDTA-containing D-PBS, centrifuged at 190$g$ for 5 min and suspended in 4 ml of the assay buffer described for the GloSensor assay. The cell suspension was dispensed into a white 96-well plate at a volume of 80 µl per well (hereafter 96-well plate) and loaded with 20 µl of 50 µM coelenterazine (Carbosynth) diluted in the assay buffer. After 2 h of incubation at room temperature, the plate was measured for baseline luminescence (SpectraMax L equipped with 2PMT, Molecular Devices; SoftMax Pro (v.7.03), Molecular Devices) and 20 µl of 6× ligand diluted in the assay buffer or vehicle was manually added. The plate was read for 15 min with a 40 s interval at room temperature. The luminescence counts from 13 to 15 min after ligand addition were averaged and normalized to the initial counts. the response were fitted to all data using the same procedure as described for the GloSensor assay.

### Flow cytometry analysis

Cell surface expression of PTH1R was measured using a previously described flow cytometry method[17]. In brief, HEK293A cells were seeded in a 6-well culture plate at a concentration of $2 \times 10^5$ cells per ml 1 day before transfection. Transfection was performed using the same procedure as described for the GloSensor assay. One day after transfection, the cells were collected by adding 100 µl of 0.53 mM EDTA-containing D-PBS, then 100 µl of 5 mM HEPES (pH 7.4) containing HBSS. The cell suspension was transferred into a 96-well V-bottom plate and fluorescently labelled with an anti-Flag epitope (DYKDDDDK) tag monoclonal antibody (Clone 1E6, FujiFilm Wako Pure Chemicals; 10 µg ml$^{-1}$ diluted in 2% goat serum and 2 mM EDTA-containing D-PBS (blocking buffer) and a goat anti-mouse IgG (H+L) secondary antibody conjugated with Alexa Fluor 488 (1:200) (ThermoFisher Scientific, 10 µg ml$^{-1}$ diluted in the blocking buffer). After washing with D-PBS, the cells were resuspended in 100 µl of 2 mM EDTA-containing D-PBS and filtered through a 40-µm filter. The fluorescence intensity of single cells was quantified using a flow cytometer (EC800 equipped with a 488 nm laser, Sony). The fluorescent signal derived from Alexa Fluor 488 was recorded in a FL1 channel, and the flow cytometry data were analysed using the FlowJo 10 software (FlowJo). Live cells were gated with a forward scatter (FS-Peak-Lin) cutoff at the 390 setting, with a gain value of 1.7. Values of mean fluorescence intensity from approximately 20,000 cells per sample were used for analysis.

### Live-cell imaging by confocal microscopy

Transfection was performed using a mixture of 500 ng mVenus–β-arrestin 2 and 200 ng Flag–PTH1R plasmids (per well in a 6-cm dish). After incubation for 1 day, the transfected cells were collected and reseeded on a 35-mm, collagen-coated glass bottom dish (Matsunami). After 1 day, medium was changed to DMEM without phenol-red and FBS (starvation buffer). After 1 h of incubation, the cells were incubated with Alexa-647-labelled (1:2,000) Flag-M1 antibody for 1 h, washed once in the starvation buffer and set on the confocal microscope. Live-cell imaging was performed using LSM880 with Airyscan (Zeiss) equipped with a ×100/1.46 alpha-Plan-Apochromat oil-immersion lens and ImmersolTM 518F/37 °C (444970-9010-000, Zeiss). During live-cell imaging, the dish was mounted in a chamber (STXG-WSKMX-SET, TOKAI HIT) to maintain the incubation conditions at 37 °C and 5% CO$_2$. We took dual-colour time-lapse images with the following settings: time interval of 5 min; total time of 35 min. The 200 µl ligand solution in 0.01% BSA-HBSS was added between time points 1 and 2. Acquired serial images were Airyscan processed using Zeiss ZEN 2.3 SP1 FP3 (black, 64 bit) (v.14.0.21.201). Co-localization analysis was performed using Fiji (v.2.0.0-rc-69/1.52p).

## Reporting summary

Further information on research design is available in the Nature Portfolio Reporting Summary linked to this article.

## Data availability

Atomic coordinates for the PCO371-bound PTH1R–mini-G$_s$β$_1$γ$_2$–Nb35 complex have been deposited into the PDB under accession code 8GW8. The associated electron microscopy data have been deposited into the Electron Microscopy Data Bank under accession code EMD-34305. All other data are provided with this paper. Source data are provided with this paper.

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

**Acknowledgements** We thank R. Danev and M. Kikkawa for setting up the cryo-EM infrastructure; K. Ogomori for technical assistance; K. Sato, S. Nakano and A. Inoue at Tohoku University for their assistance in plasmid preparation and the cell-based GPCR assays; and K. Yamashita for model validation. This work was supported by JSPS KAKENHI grants JP20J21820 (K. Kobayashi), JP21H05037 (O.N.), JP21H04791 (A.I.), JP21H05113 (A.I.), JP18H05425 (T.M), JPJSBP120213501 (A.I.) and JPJSBP120218801 (A.I.); JP19gm5910013, JP20gm0010004, JP20am0101095, JP22ama121038 and JP22zf0127007 (A.I.) from the Japan Agency for Medical Research and Development (AMED); JP22am0101115, JP22ama121002 (support number 3272), JP22ama121012 (support number 4826) and JP233fa627001 (O.N.); JPMJFR215T and JPMJMS2023 (A.I.) from the Japan Science and Technology Agency (JST); JST CREST programme 20344981 (O.N.); Takeda Science Foundation (A.I.); the Uehara Memorial Foundation (A.I.); Tokyo Biochemical Research Foundation (A.I.); and Daiichi Sankyo Foundation of Life Science (A.I.); and a research grant from Chugai Pharmaceutical (A.I., T.M. and O.N.). Molecular graphics and analyses were performed using UCSF ChimeraX 1.0, developed by the Resource for Biocomputing, Visualization, and Informatics at the University of California, San Francisco, with support from the National Institutes of Health (R01-GM129325) and the Office of Cyber Infrastructure and Computational Biology, National Institute of Allergy and Infectious Diseases.

**Author contributions** K. Kobayashi designed the entire experiment. K. Kobayashi. K.S. and H.H.O. constructed and purified the plasmids. K. Kobayashi purified the mini-G$_s$ heterotrimer, Nb35 and PCO371-bound PTH1R, and prepared the complex and cryo-EM grids. K. Kobayashi, K. Kawakami, S.H. and H.N. performed the mutation assays and analyses. H.M. and M.S. performed analyses and directed this programme based on their biological expertise. K. Kobayashi and T.K. collected cryo-EM data, and K. Kobayashi processed the cryo-EM data. K. Kobayashi built models and performed modelling and refinement. A.T. designed the system of molecular dynamics simulation. A.T. and K. Kobayashi performed and analysed the molecular dynamics simulation. K. Kobayashi prepared the initial manuscript, and K. Kobayashi, K. Kawakami, A.T., T.K. and M.N. wrote the manuscript with input from all authors. T.K., A.I., T.M. and O.N. supervised the research.

**Competing interests** O.N. is a co-founder and an external director of Curreio Inc. H.N., H.M. and M.S. are employees of Chugai Pharmaceutical. All other authors declare no competing interests.

## Additional information

**Correspondence and requests for materials** should be addressed to Asuka Inoue, Takeshi Murata or Osamu Nureki.

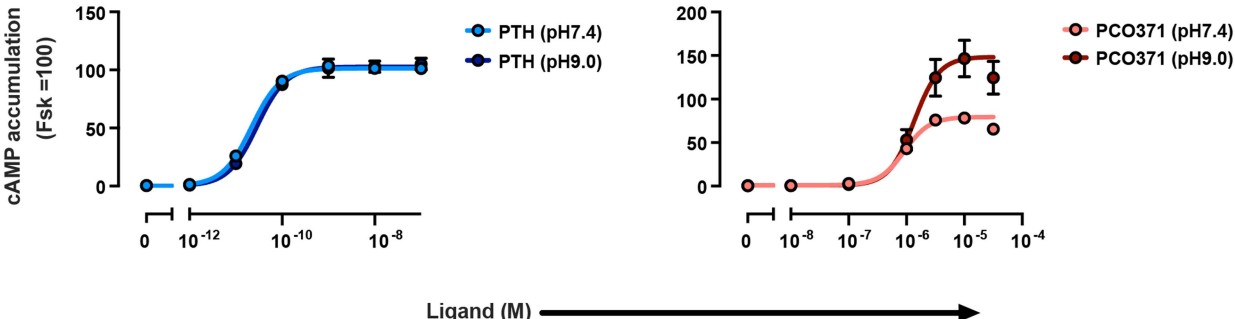

**Extended Data Fig. 1 | PCO371 and PTH-induced PTH1R activation under two different conditions.** pH sensitivity of the GloSensor cAMP responses of PTH1R upon stimulation with PTH and PCO371. Note that we adjusted the assay buffer (See also Methods) to the indicated pH. Symbols and error bars represent mean and SEM, respectively, of three independent experiments with each performed in duplicate.

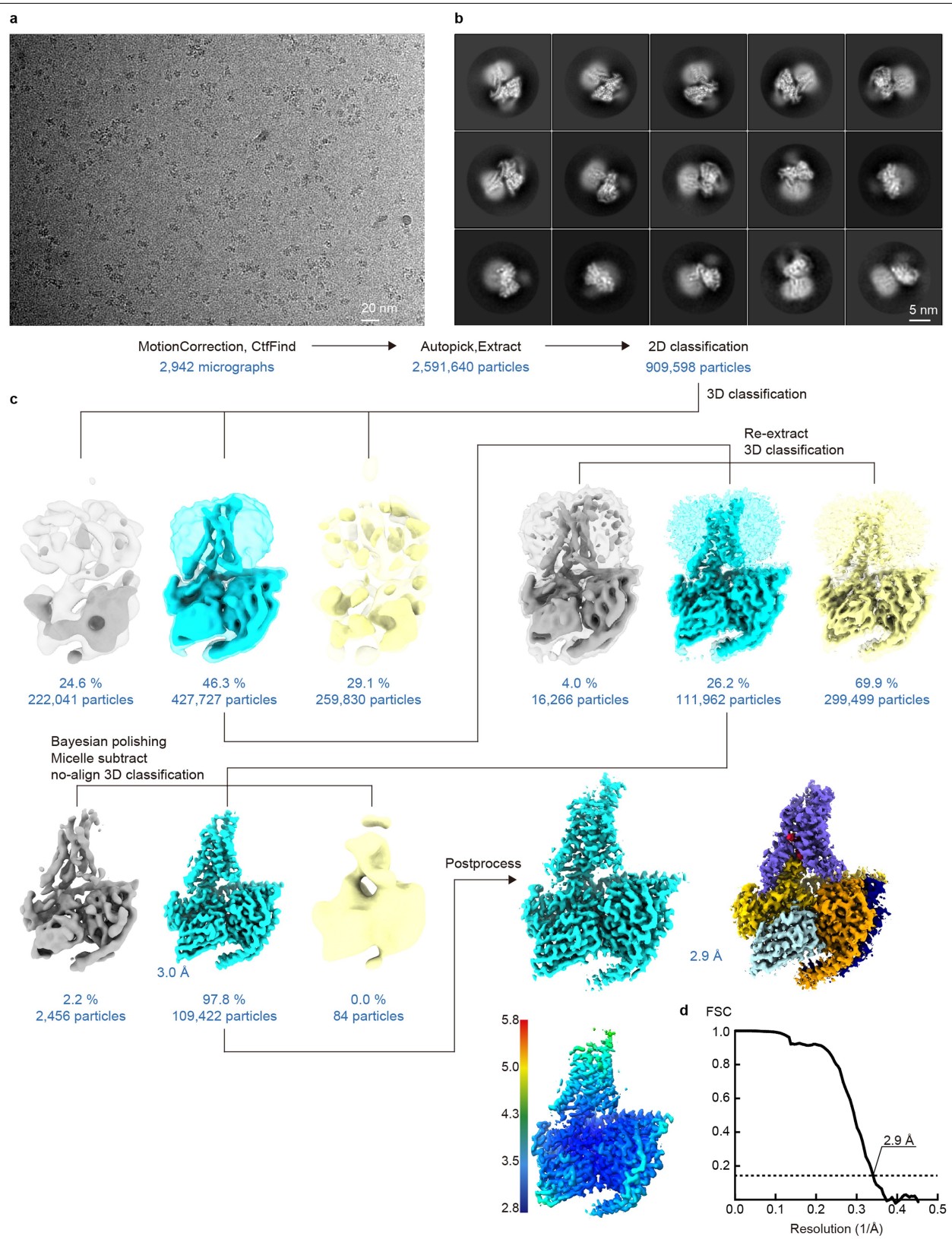

**Extended Data Fig. 2 | Cryo-EM data processing of the PCO371–PTH1R–G$_s$ complex.** (a) Representative micrograph from 6,333 images of PCO371–PTH1R–G$_s$ data set, showing the distribution of PCO371–PTH1R–G$_s$ particles. (b) Representative two-dimensional class averages, indicating secondary structure features. The diameter of the circular windows is 18 nm. (c) Data processing workflow for the PCO371–PTH1R–G$_s$ complex after 3D classification. (d) Gold-standard Fourier shell correlation (FSC) curves, indicating overall global resolution at 2.9 Å.

**a**

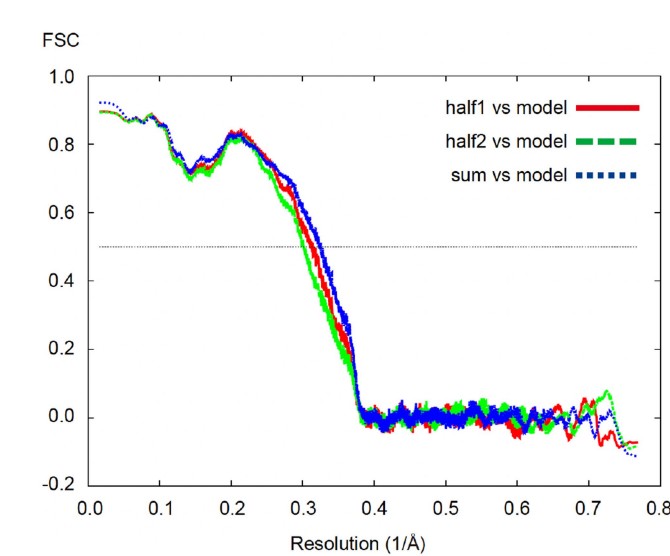

TM1  TM2  TM3  TM4  TM5  TM6  TM7

ICL-1  ICL-2  ICL-3  Helix8  α-5 Helix

45°

PCO371

**b**

FSC

half1 vs model
half2 vs model
sum vs model

Resolution (1/Å)

**Extended Data Fig. 3 | Model of PCO371–PTH1R–G$_s$ in the cryo-EM density map.** (a) Cryo-EM density maps and models for PTH1R and PCO371. (b) Cross-validation of each refined model using two half maps. Overfitting of refined models was tested with a previously described cross-validation method[53].

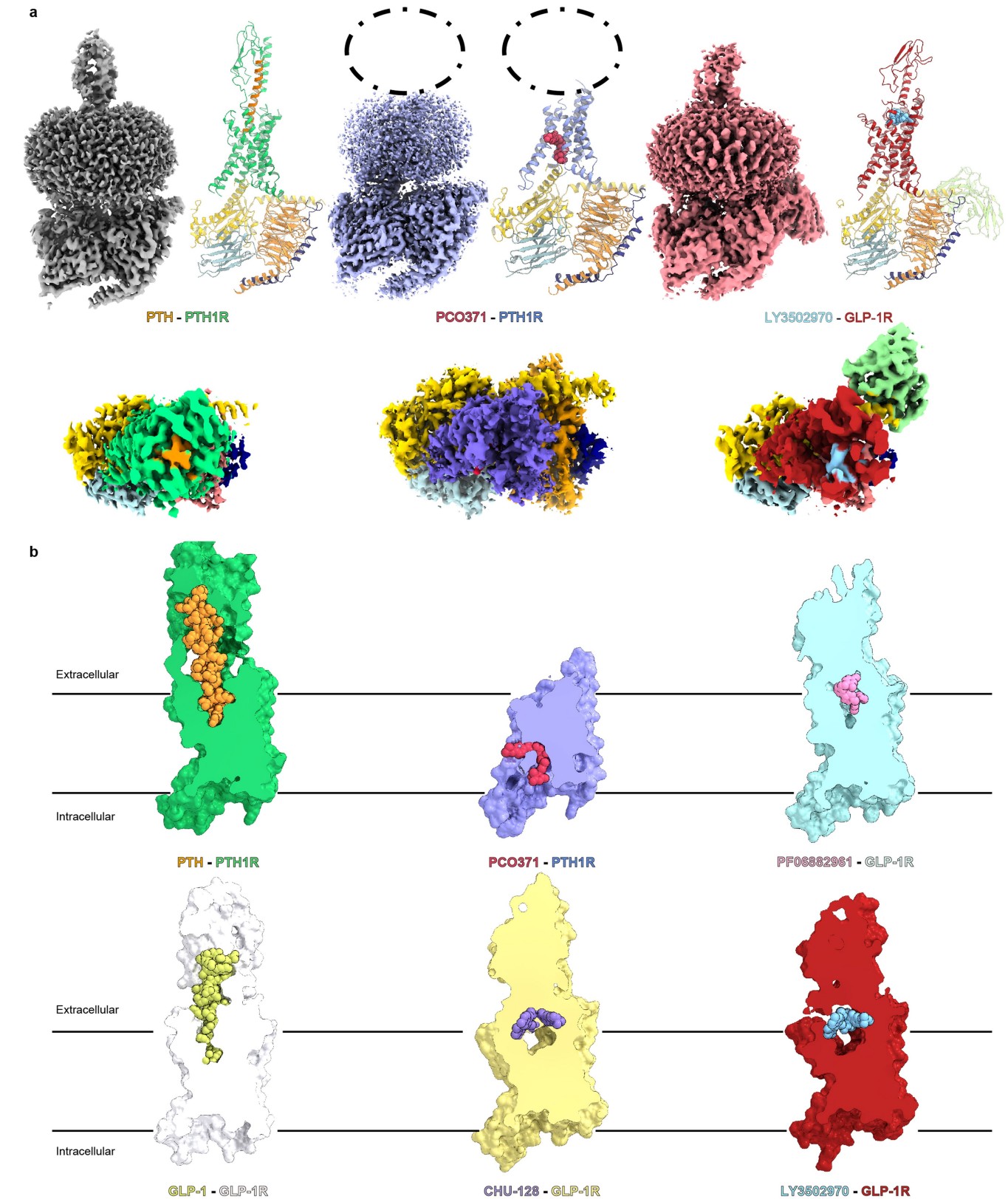

**Extended Data Fig. 4** | See next page for caption.

**Extended Data Fig. 4 | Comparison of the PCO371–PTH1R–G$_s$ structure with the endogenous and chemical agonists-bound PTH1R and GLP-1R structures.** (a) Comparison of the PCO371–PTH1R–G$_s$ complex with endogenous peptide-bound and orthosteric chemical agonist-bound class B1 GPCR complexes. Because there is no structure of orthosteric chemical agonist-bound PTH1R structure, we used LY3502970 (a proprietary GLP-1 receptor agonist)–GLP-1R–G$_s$ to represent the structures of orthosteric chemical agonist-bound class B1 GPCRs. Orthogonal views and models show PTH–PTH1R–G$_s$ (PDB: 7VVL), PCO371–PTH1R–G$_s$ (8GW8), and LY3502970–GLP-1R–G$_s$ (PDB: 6XOX). Green, PTH-bound PTH1R; orange, PTH; yellow, mini-G$_s$ Ras-like domain; tomato, Gβ$_1$; navy, Gγ$_2$; powder blue, Nb35; violet, PCO371-bound PTH1R; crimson, LY3502970-bound GLP-1R; sky blue, LY3502970. Gray, PTH–PTH1R–G$_s$; violet, PCO371–PTH1R–G$_s$; red, LY3502970–GLP-1R–G$_s$. The PCO371-bound PTH1R map shows no obvious ligand-like density in the orthosteric pocket, in contrast to the PTH-bound PTH1R and LY3502970-bound GLP-1R maps. (b) Cut-through views of endogenous peptide-bound PTH1R and GLP-1R structures and chemical agonist-bound GLP-1R structures. Green, PTH-bound PTH1R; orange, PTH; white, GLP-1R; yellow-green, GLP-1; violet, PCO371-bound PTH1R; crimson, PCO371; yellow, CHU-128-bound GLP-1R; blue-violet, CHU-128; purple, danuglipron (PF-06882961)-bound GLP-1R; khaki, PF-06882961; red, LY3502970-bound GLP-1R; sky blue, LY3502970. Only PCO371 binds within the intracellular cavity, whereas other agonists bind in various ways to the orthosteric pocket.

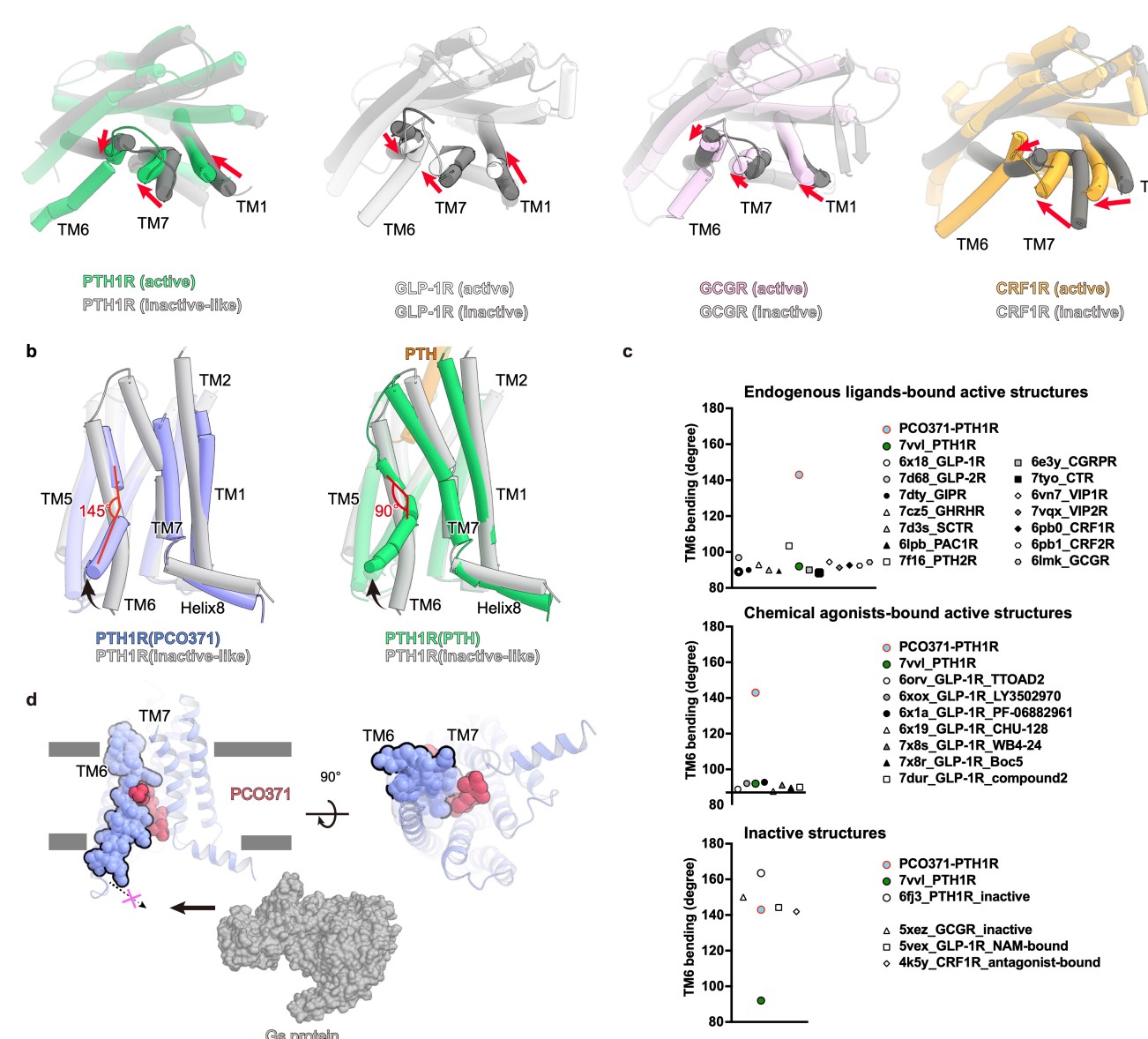

**Extended Data Fig. 5 | Comparison of the PCO371-bound PTH1R conformation with other class B1 GPCR structures.** (a) Four active class B1 GPCRs (green, PTH1R; white, GLP-1R; pink, glucagon receptor [GCGR]; purple, corticotropin releasing factor receptor 1 [CRF1R]) are superimposed onto inactive or inactive-like structures. (b) PTH-bound PTH1R and PCO371-bound PTH1R are superimposed onto the ePTH-bound inactive-like PTH1R structure (green, PTH-bound PTH1R; orange, PTH; violet, PCO371-bound PTH1R;

magenta, PCO371; gray, ePTH-bound PTH1R). PCO371-bound PTH1R exhibited the moderate kink in TM6 (approximately 145°), whereas PTH-bound PTH1R exhibited the typical sharp kink in TM6 (approximately 90°). (c) TM6 angle consisting of the three Cα atoms of the I/M/V$^{6.39}$, G$^{6.50}$, and A/V/L$^{6.54}$ residues. (d) PCO371 directly stabilizes the intracellular half of TM6 in the outward conformation, which increases the volume in the inner cavity and activates the G$_s$ protein.

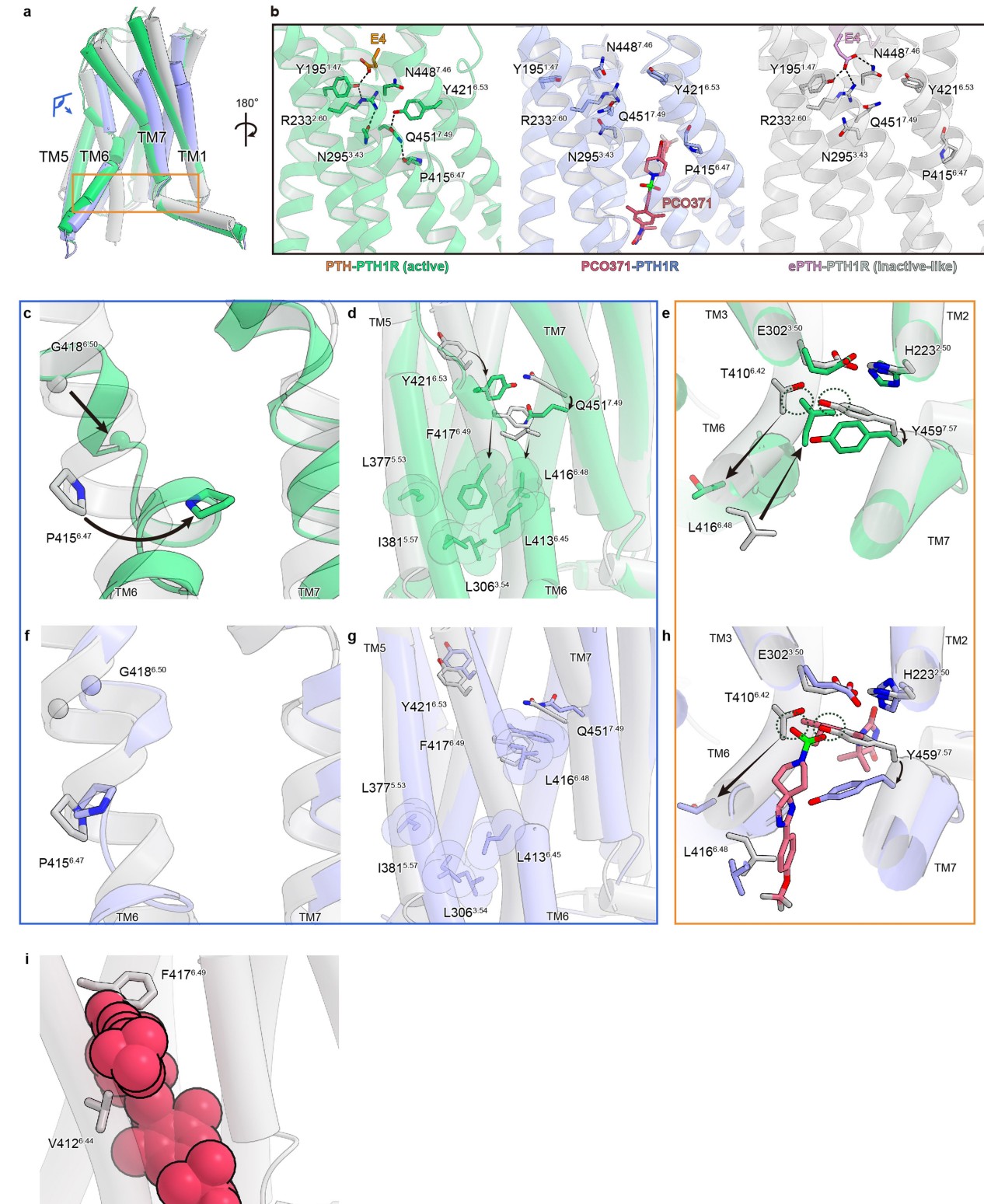

**Extended Data Fig. 6 | Comparison of the PxxG switch and the HETY inactive motif between PCO371-bound and PTH-bound PTH1R.** (a) TMD region of the PCO371-bound PTH1R. The eye and square indicate views in (c–h). (b) Central polar networks of bound PTH-bound active PTH1R (left, violet), PCO371-PTH1R (center, green), and ePTH-bound inactive PTH1R (right, gray).

(c–h) Superimposed structures of PTH-bound PTH1R (green), PCO371-PTH1R (violet), and ePTH-bound PTH1R (gray) are shown parallel to the membrane. Magnified views of the PYQ active motif and the PxxG active switch (c–d and f–g). Magnified views of the HETY inactive motif (e and h). (i) Superimposed structures of PCO371 (crimson) and ePTH-bound PTH1R.

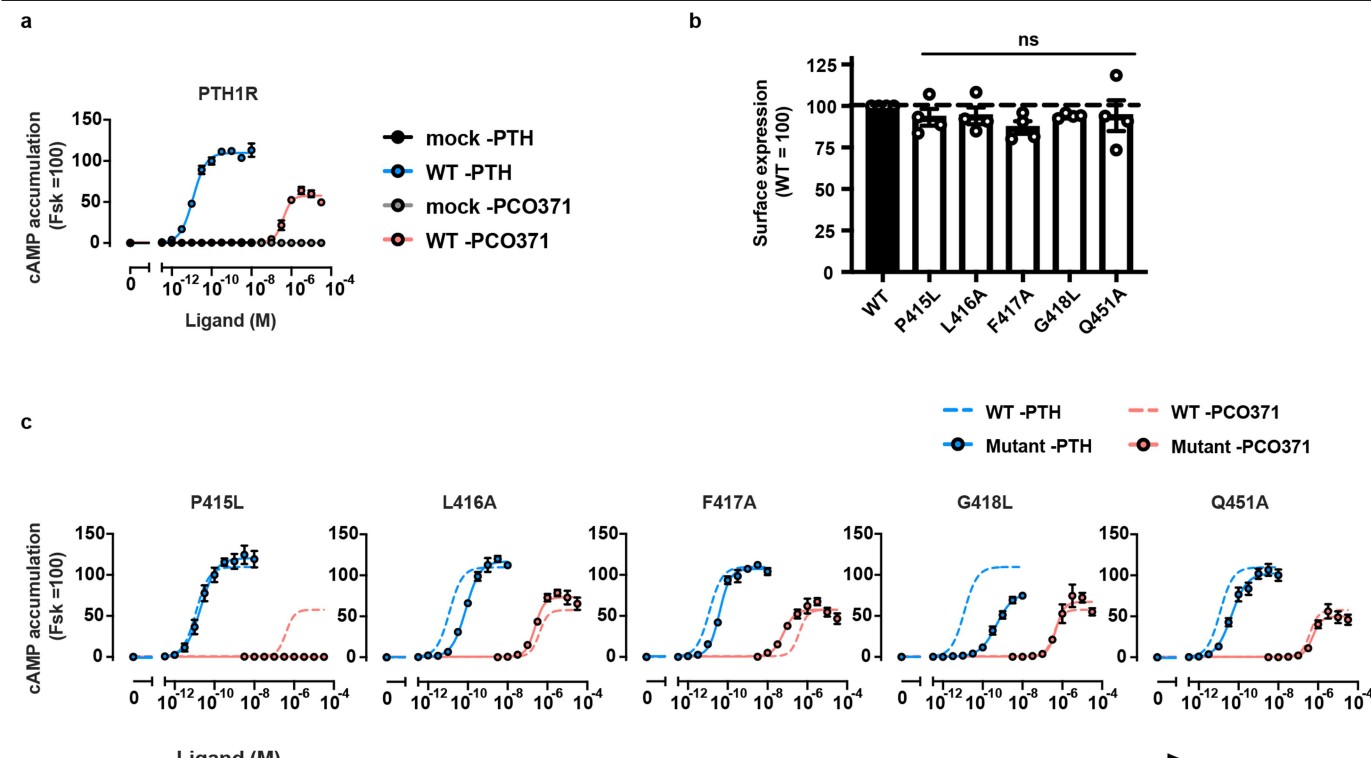

**Extended Data Fig. 7 | GloSensor cAMP responses and surface expression of wild-type and mutant PTH1R.** Related to Fig. 3d. (a–c) PTH1R-induced $G_s$ activation measured by the GloSensor cAMP accumulation assay and cell surface expression of WT and mutant PTH1R assessed by the flow cytometry analysis. (a) Concentration-response curves of the GloSensor cAMP response in WT PTH1R- or mock-transfected cells upon PTH or PCO371 stimulation. Symbols and error bars represent mean and SEM, respectively, of three independent experiments with each performed in duplicate. (b) Cell surface expression. Symbols and error bars represent mean and SEM, respectively, and dots show individual data of four independent experiments, with each performed in duplicate. Statistical analysis was performed by one-way ANOVA followed by Dunnett's test for multiple comparison analysis (with reference to the WT). ns, not significantly different between the groups. (c) Concentration-response curves of GloSensor cAMP response of WT (dashed lines) and the mutant (solid lines) PTH1R upon PTH (blue) or PCO371 (red) stimulation. Symbols and error bars represent mean and SEM, respectively, of three independent experiments with each performed in duplicate.

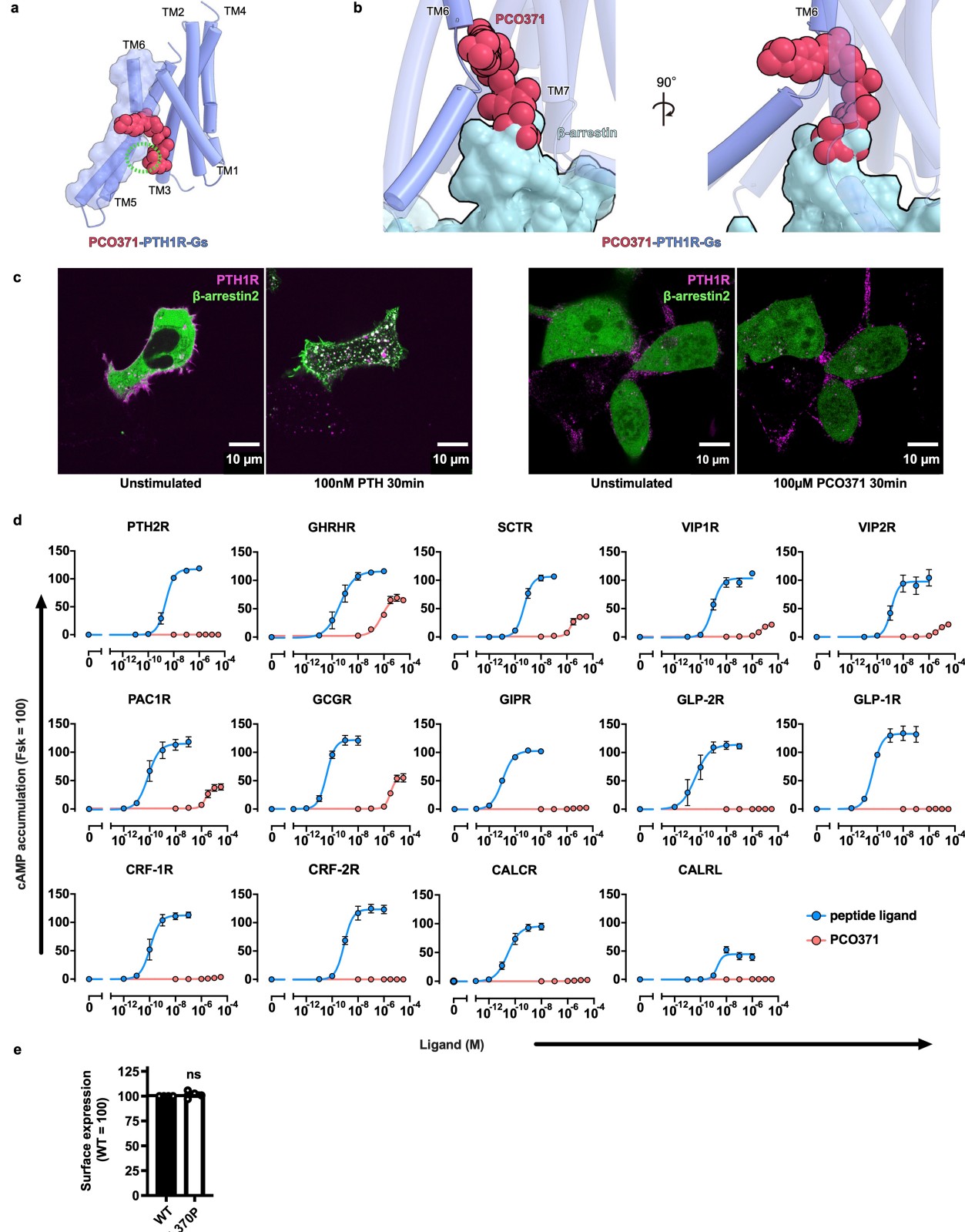

**Extended Data Fig. 8** | See next page for caption.

**Extended Data Fig. 8 | PCO371-induced activation of class B1 GPCRs.** (a) The TMD structures of PCO371-bound PTH1R (violet) and PCO371 (crimson) are shown parallel to the membrane. (b) Superimposed structures of PCO371–PTH1R–$G_s$ and formoterol–$\beta_1AR$–$\beta$-arrestin complexes, aligned on TMs 2-5. PCO371 facilitates the outward conformation of the intracellular portion of TM6 and clashes with $\beta$-arrestin. (c) Representative images of the cellular localization of Alexa-647-labeled PTH1R (magenta) and mVenus-fused $\beta$-arrestin 2 (green), related to Fig. 3g. These images obtained from 10-19 images of these experiments. (d) Concentration-response curves of the GloSensor cAMP response of 15 class B1 GPCRs upon endogenous peptide agonist (blue) or PCO371 (red) stimulation. The following endogenous peptide agonists were used for each GPCR; PTH (1–34), PTH2R: Growth hormone-releasing hormone, GHRHR; Secretin, SCTR; Pituitary Adenylate Cyclase Activating Polypeptide (PACAP, 1–27), VIP1R and VIP2R; PACAP (1–38), PAC1R; Glucagon, GCGR; Gastric Inhibitory Polypeptide, GIPR; Glucagon-like Peptide 2, GLP-2R; Glucagon-like Peptide 1, GLP-1R; Corticotropin Releasing Factor, CRF1R and CRF2R; Calcitonin, CALCR; Calcitonin Gene Related Peptide, CALRL. All peptides are human-derived sequence. Symbols and error bars represent mean and SEM, respectively, of three independent experiments with each performed in duplicate. (e) Cell surface expression of WT and mutant PTH2R assessed by the flow cytometry analysis. Symbols and error bars represent mean and SEM, respectively, and dots show individual data of four independent experiments, with each performed in duplicate. Statistical analysis was performed by two-tailed t-test. ns, not significantly different between the groups.

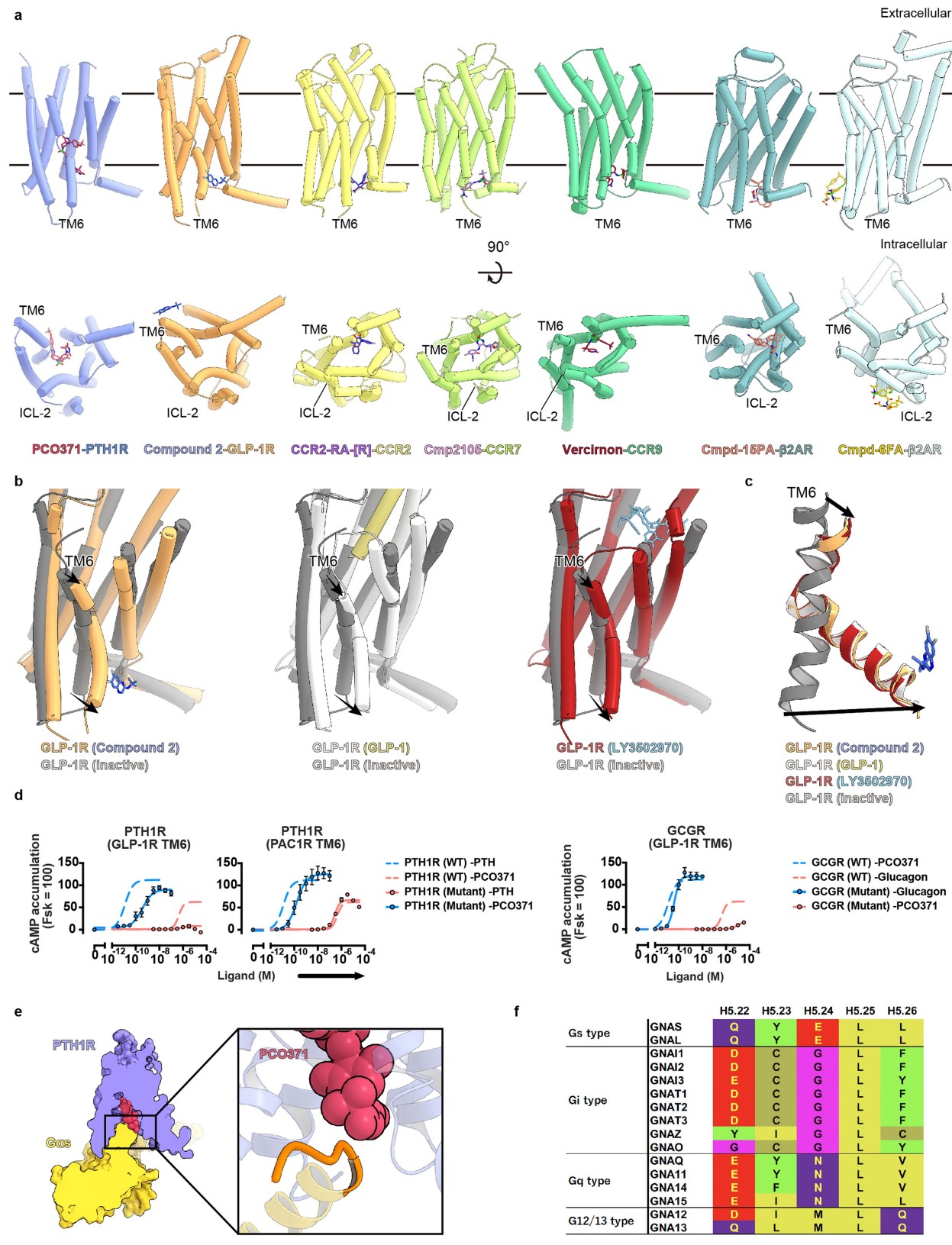

**Extended Data Fig. 9** | See next page for caption.

**Extended Data Fig. 9 | Structural comparison of PCO371-bound PTH1R and the other intracellular ligands-bound GPCR structures.** (a) TMD structures of PCO371-bound PTH1R and six intracellular ligand-bound GPCRs are shown parallel to the membrane or from the intracellular side. Violet, PCO371-bound PTH1R; magenta, PCO371; orange, compound 2-bound GLP-1R; blue, compound 2, yellow, C-C motif chemokine receptor (CCR2)-RA-[R]-bound CCR2; violet, CCR2-RA-[R]; yellow-green, cmpd2105-bound CCR7; plum, cmpd2105; green, vercirnon-bound CCR6; brown, vercirnon; powder blue, cmpd-15PA-bound $\beta_2$AR; coral, cmpd-15PA; light cyan, cmpd-6FA-bound $\beta_2$AR; gold, cmpd-6FA. (b, c) Superimposed structures of compound2-bound GLP-1R, GLP-1-bound GLP-1R, LY3502970-bound GLP-1R, and inactive GLP-1R. White,

GLP-1-bound GLP-1R; khaki, GLP-1; red, LY3502970 -bound GLP-1R; sky blue, LY3502970; gray, inactive GLP-1R. Compound2 binds with the intracellular portion of TM6, and compound2-bound GLP-1R exhibits the identical active conformation of TM6. (d) GloSensor cAMP responses in wild-type (WT) PTH1R and the TM6-replaced mutants upon endogenous agonists (PTH or glucagon) or PCO371 stimulation. Symbols and error bars represent mean and SEM, respectively, of three independent experiments with each performed in duplicate. (e) Cut-through view of PCO371–PTH1R–$G_s$ and magnified view of the PCO371-G-protein interface. The PCO371-interacting $G\alpha_s$ region is colored orange. (f) Sequence alignment of the C-terminal hook of $G\alpha$ proteins.

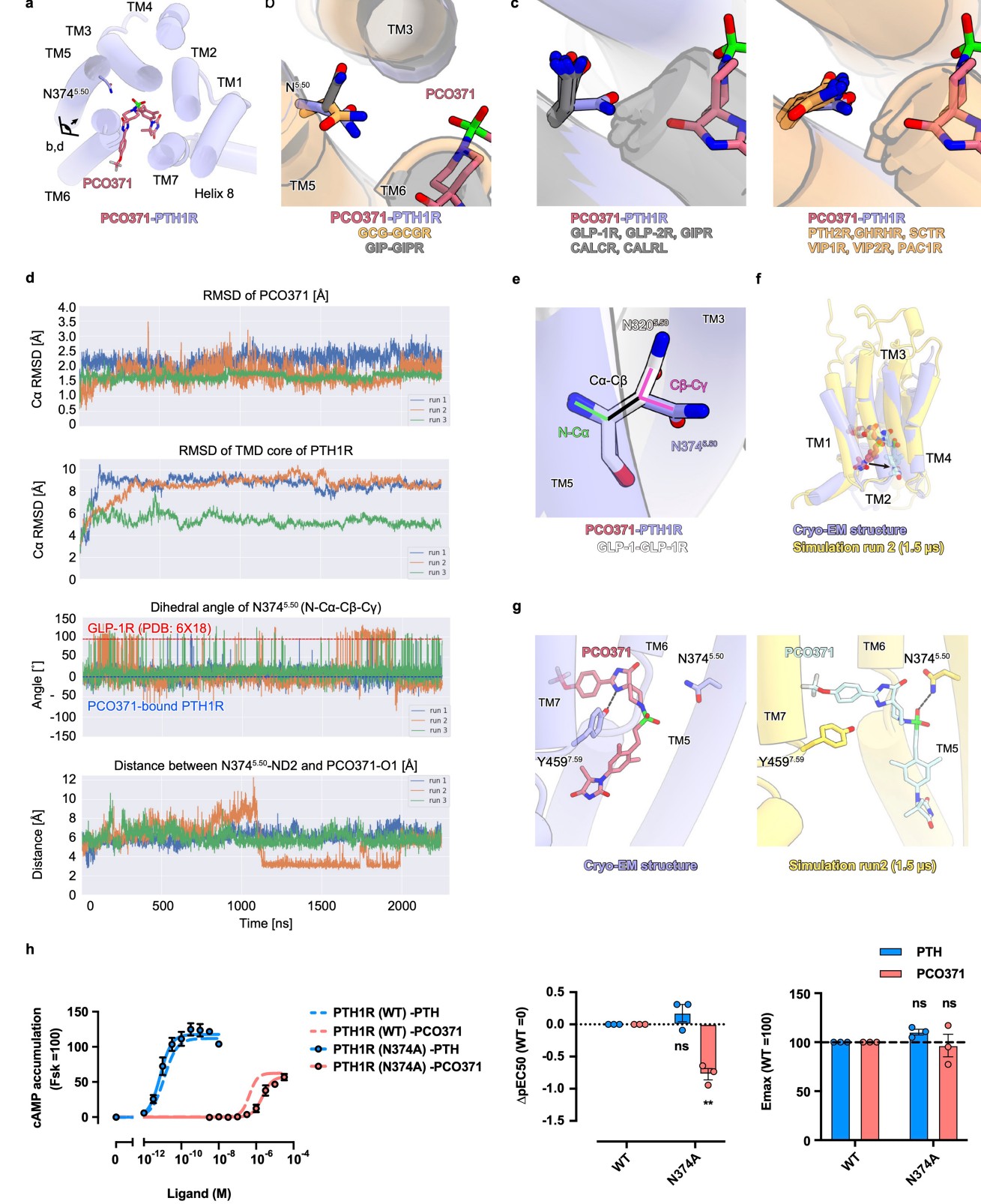

**Extended Data Fig. 10** | See next page for caption.

**Extended Data Fig. 10 | MD simulation from the PCO371-bound PTH1R.**
(a) A cross-section view of the PCO371-bound PTH1R. (b) A magnified view of the superimposed structure of PCO371-bound PTH1R (purple), GCG-bound GCGR (orange), and GIP-bound GIPR (gray). (c) Structural comparison of $N^{5.50}$ of PCO371-insensitive receptors and PCO371-sensitive receptors, related to Fig. 4d. Violet, PCO371-bound PTH1R; gray, PCO371-insensitive receptors; orange, PCO371-sensitive receptors. (d) Trajectory analysis of three independent simulations with PCO371-bound PTH1R. The top two panels show that PCO371-bound PTH1R maintains an active conformation, resembling our cryo-EM structure. The panel second from the bottom shows the dihedral angle of $N374^{5.50}$. The angles of PCO371-bound PTH1R (blue) and GLP-1R (red) are shown as dash lines. The bottom panel shows the distance between $N374^{5.50}$-ND2 and PCO371-O1, representing stably PCO371 interaction with $N374^{5.50}$ in an intermediate state. (e) Distinct conformation of $N374^{5.50}$ between PCO371-bound PTH1R and GLP-1-bound active GLP-1R (PDB ID: 6X18). The dihedral angle is calculated with the N–Cα and Cβ–Cγ angles of $N^{5.50}$. (f) Superimposition of our cryo-EM structure (violet) and a representative snapshot of 1.5 μs MD simulation in run 2 of PCO371-bound PTH1R (yellow). In the simulation, the receptor adopted an intermediate structure, and PCO371 moved toward TM5. (g) Comparison of our cryo-EM structure (violet) and an intermediate structure (yellow) observed in the simulation. PCO371 does not interact with $N374^{5.50}$ in our cryo-EM structure, while it creates a stable interaction with $N374^{5.50}$ in the simulation. (h) cAMP accumulation of WT and $N374^{5.50}$A mutant with PTH and PCO371. The $N374^{5.50}$A mutant selectively reduced PCO371 response. Symbols and error bars represent mean and SEM, respectively, and dots show individual data of three independent experiments, with each performed in duplicate. * and ** represent $P < 0.05$ and $0.01$, respectively, with two-way ANOVA followed by Dunnett's test for multiple comparison analysis. ns, not significantly different between the groups.

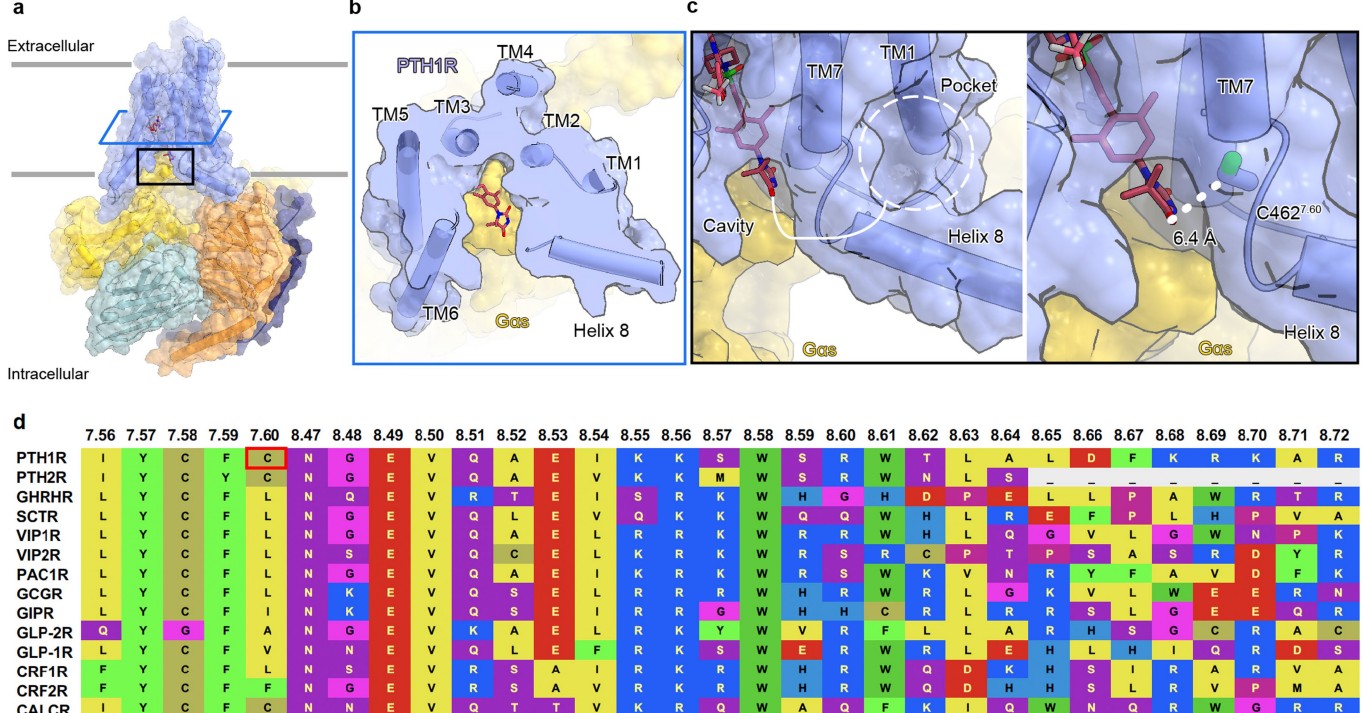

**Extended Data Fig. 11 | Proposed strategy for a G-protein subtype specific or bitopic agonist.** (a) TMD structures of PCO371–PTH1R–G$_s$ are shown parallel to the membrane. (b, c) Cross-section and magnified views of the TMD. The white dashed circle shows an intracellular ligand binding pocket comprising TM1, TM7, and Helix 8 (left panel). The white dashed line shows the distance between Cβ of Cys[7.60] and O6 of PCO371. (d) Sequence alignment of the C-terminal region after TM7. The red box shows C462[7.60] of PTH1R.

| | 7.56 | 7.57 | 7.58 | 7.59 | 7.60 | 8.47 | 8.48 | 8.49 | 8.50 | 8.51 | 8.52 | 8.53 | 8.54 | 8.55 | 8.56 | 8.57 | 8.58 | 8.59 | 8.60 | 8.61 | 8.62 | 8.63 | 8.64 | 8.65 | 8.66 | 8.67 | 8.68 | 8.69 | 8.70 | 8.71 | 8.72 |
|---|---|---|---|---|---|---|---|---|---|---|---|---|---|---|---|---|---|---|---|---|---|---|---|---|---|---|---|---|---|---|---|
| PTH1R | I | Y | C | F | C | N | G | E | V | Q | A | E | I | K | K | S | W | S | R | W | T | L | A | L | D | F | K | R | K | A | R |
| PTH2R | I | Y | C | Y | C | N | G | E | V | Q | A | E | V | K | K | M | W | S | R | W | N | L | S | | | | | | | | |
| GHRHR | L | Y | C | F | L | N | Q | E | V | R | T | E | I | S | R | K | W | H | G | H | D | P | E | L | L | P | A | W | R | T | R |
| SCTR | L | Y | C | F | L | N | G | E | V | Q | L | E | V | Q | K | K | W | Q | Q | W | H | L | R | E | F | P | L | H | P | V | A |
| VIP1R | L | Y | C | F | L | N | G | E | V | Q | A | E | L | R | R | K | W | R | R | W | H | L | Q | G | V | L | G | W | N | P | K |
| VIP2R | L | Y | C | F | L | N | S | E | V | Q | C | E | L | K | R | K | W | R | S | R | C | P | T | P | S | A | S | R | D | Y | R |
| PAC1R | L | Y | C | F | L | N | G | E | V | Q | A | E | I | K | R | K | W | R | S | W | K | V | N | R | Y | F | A | V | D | F | K |
| GCGR | L | Y | C | F | L | N | K | E | V | Q | S | E | L | R | R | W | H | R | W | R | L | G | K | V | L | W | E | E | R | N | |
| GIPR | L | Y | C | F | I | N | K | E | V | Q | S | E | I | R | G | W | H | H | C | R | L | R | R | S | L | G | E | E | Q | R | |
| GLP-2R | Q | Y | G | F | A | N | G | E | V | K | A | E | L | R | K | Y | W | V | R | F | L | L | A | R | H | S | G | C | R | A | C |
| GLP-1R | L | Y | C | F | V | N | N | E | V | Q | L | E | F | R | K | S | W | E | R | W | R | L | E | H | L | H | I | Q | R | D | S |
| CRF1R | F | Y | C | F | L | N | S | E | V | R | S | A | I | R | K | W | H | R | W | Q | D | K | H | S | I | R | A | R | V | A | |
| CRF2R | F | Y | C | F | F | N | G | E | V | R | S | A | V | R | K | W | H | R | W | Q | D | H | H | S | L | R | V | P | M | A | |
| CALCR | I | Y | C | F | C | N | N | E | V | Q | T | T | V | K | R | Q | W | A | Q | F | K | I | Q | W | N | Q | R | W | G | R | R |
| CALRL | I | F | C | F | F | N | G | E | V | Q | A | I | L | R | R | N | W | N | Q | Y | K | I | Q | F | G | N | S | F | S | N | S |

**Extended Data Table 1 | Cryo-EM data collection and refinement statistics**

| | #1 PCO371–PTH1R–Gs complex (PDB ID 8GW8, EMD-34305) |
|---|---|
| **Data collection and processing** | |
| Magnification | 105,000 |
| Voltage (kV) | 300 |
| Electron exposure (e⁻/Å²) | 54.578 |
| Defocus range (μm) | 0.8~1.6 |
| Pixel size (Å) | 0.83 |
| Symmetry imposed | C1 |
| Initial particle images (no.) | 2,591,640 |
| Final particle images (no.) | 109,422 |
| Map resolution (Å) | 2.9 |
| Map resolution range (Å) | 2.8 -5.8 |
| Map sharpening $B$ factor (Å²) | -80.106 |
| | |
| **Refinement** | |
| Initial model used (PDB code) | 7VVL |
| RMSD from ideal | |
| Bond lengths (Å) | 0.0026 |
| Bond angles (°) | 0.53 |
| | |
| **Validation** | |
| MolProbity score | 1.52 |
| Clashscore | 6.49 |
| Poor rotamers (%) | 0.24 |
| Ramachandran plot | |
| Favored (%) | 97.07 |
| Allowed (%) | 2.93 |
| Outliers (%) | 0.00 |

**Extended Data Table 2 | Observed intermolecular interactions of PCO371 within the TMD of receptor and Gα$_s$**

| PCO371 | Receptor | Gαs | Interaction |
|---|---|---|---|
| Trifluoromethoxyphenyl | L411 | | |
| | F416 | | |
| | F454 | | Hydrophobic interaction |
| | I458 | | |
| Spiro-imidazolone | L226 | | |
| | I299 | | |
| | E302 | | |
| | M414 | | Hydrophobic interaction |
| | P415 | | |
| | F417 | | |
| | Y459 | | Hydrogen bond |
| Dimethylphenyl | H223 | | Hydrogen bond |
| | L306 | | Salt bridge |
| | L413 | | Hydrophobic interaction |
| Dimethylhydantoin | R219 | | Salt bridge |
| | V412 | | Hydrophobic interaction |
| | N463 | | Hydrogen bond |
| | | Y391 | |
| | | E392 | Hydrophobic interaction |
| | | L393 | |

# Reporting Summary

## Statistics

For all statistical analyses, confirm that the following items are present in the figure legend, table legend, main text, or Methods section.

| n/a | Confirmed | |
|---|---|---|
| ☐ | ☒ | The exact sample size (*n*) for each experimental group/condition, given as a discrete number and unit of measurement |
| ☐ | ☒ | A statement on whether measurements were taken from distinct samples or whether the same sample was measured repeatedly |
| ☐ | ☒ | The statistical test(s) used AND whether they are one- or two-sided<br>*Only common tests should be described solely by name; describe more complex techniques in the Methods section.* |
| ☒ | ☐ | A description of all covariates tested |
| ☐ | ☒ | A description of any assumptions or corrections, such as tests of normality and adjustment for multiple comparisons |
| ☐ | ☒ | A full description of the statistical parameters including central tendency (e.g. means) or other basic estimates (e.g. regression coefficient) AND variation (e.g. standard deviation) or associated estimates of uncertainty (e.g. confidence intervals) |
| ☐ | ☒ | For null hypothesis testing, the test statistic (e.g. *F*, *t*, *r*) with confidence intervals, effect sizes, degrees of freedom and *P* value noted<br>*Give P values as exact values whenever suitable.* |
| ☒ | ☐ | For Bayesian analysis, information on the choice of priors and Markov chain Monte Carlo settings |
| ☒ | ☐ | For hierarchical and complex designs, identification of the appropriate level for tests and full reporting of outcomes |
| ☒ | ☐ | Estimates of effect sizes (e.g. Cohen's *d*, Pearson's *r*), indicating how they were calculated |

*Our web collection on statistics for biologists contains articles on many of the points above.*

## Software and code

Policy information about availability of computer code

| Data collection | Serial EM3.7.4, SoftMax Pro ver7.0.3,  Zeiss ZEN 2.3 SP1 FP3 (black, 64 bit) (ver.14.0.21.201) |
|---|---|
| Data analysis | RELION-3.1, CTFFIND4.1, COOT-0.9.3 EL,  PHENIX-1.14-3260, REFMAC5, CUEMOL-2.2.3.443, UCSF Chimera X 1.0, Prism 9, FIji (ver. 2.0.0-rc-69/1.52p), FlowJo, MODELLER 10.1, VMD 1.9.3, Charmm36, MemProtMD, CHARMM-GUI, NAMD 2.13, mdtraj-1.9.8 |

For manuscripts utilizing custom algorithms or software that are central to the research but not yet described in published literature, software must be made available to editors and reviewers. We strongly encourage code deposition in a community repository (e.g. GitHub). See the Nature Portfolio guidelines for submitting code & software for further information.

## Data

Policy information about availability of data

All manuscripts must include a data availability statement. This statement should provide the following information, where applicable:
- Accession codes, unique identifiers, or web links for publicly available datasets
- A description of any restrictions on data availability
- For clinical datasets or third party data, please ensure that the statement adheres to our policy

Atomic coordinates for the PCO371-bound PTH1R–mini–Gsβ1γ2–Nb35 complex have been deposited in the Protein Data Bank under accession code 8GW8. The associated electron microscopy data have been deposited in the Electron Microscopy Data Bank under accession code EMD-34305. All other data provided with this paper.

# Human research participants

Policy information about studies involving human research participants and Sex and Gender in Research.

| | |
|---|---|
| Reporting on sex and gender | No concern about this section in this study. |
| Population characteristics | No concern about this section in this study. |
| Recruitment | No concern about this section in this study. |
| Ethics oversight | No concern about this section in this study. |

Note that full information on the approval of the study protocol must also be provided in the manuscript.

# Field-specific reporting

Please select the one below that is the best fit for your research. If you are not sure, read the appropriate sections before making your selection.

☒ Life sciences  ☐ Behavioural & social sciences  ☐ Ecological, evolutionary & environmental sciences

For a reference copy of the document with all sections, see nature.com/documents/nr-reporting-summary-flat.pdf

# Life sciences study design

All studies must disclose on these points even when the disclosure is negative.

| | |
|---|---|
| Sample size | Sample sizes were determined based on prior literature best practices in the field; no statistical methods were used to predetermine sample size. We performed three to five independent experiments and evaluated 21-26 cells in the cell-based functional assays and the confocal microscopy analysis, respectively,  based on other similar methodologies (PMID 31160049, 35087057) |
| Data exclusions | No data was excluded from the analysis. |
| Replication | Cell-based experiments were independently performed at least three times. We verify that all experiments were successfully performed in duplicate. |
| Randomization | For cryo-EM studies, particles were randomly assigned to half-maps for resolution determination. Randomization was not relevant to the other experiments in our study as these assays don't have unknown covariates. |
| Blinding | Blinding was not relevant to the experiments in our study since no subjective allocation was involved in our study. |

# Reporting for specific materials, systems and methods

We require information from authors about some types of materials, experimental systems and methods used in many studies. Here, indicate whether each material, system or method listed is relevant to your study. If you are not sure if a list item applies to your research, read the appropriate section before selecting a response.

## Materials & experimental systems

| n/a | Involved in the study |
|---|---|
| ☐ | ☒ Antibodies |
| ☐ | ☒ Eukaryotic cell lines |
| ☒ | ☐ Palaeontology and archaeology |
| ☒ | ☐ Animals and other organisms |
| ☒ | ☐ Clinical data |
| ☒ | ☐ Dual use research of concern |

## Methods

| n/a | Involved in the study |
|---|---|
| ☒ | ☐ ChIP-seq |
| ☐ | ☒ Flow cytometry |
| ☒ | ☐ MRI-based neuroimaging |

# Antibodies

| | |
|---|---|
| Antibodies used | anti-FLAG epitope tag (DYKDDDDK) mouse monoclonal antibody (Clone 1E6, FujiFilm Wako Pure Chemical, cat. no. 012-22384); goat anti-mouse IgG secondary antibody conjugated with Alexa Fluor 488 (Thermo Fisher Scientific, cat. no. A32723); anti-FLAG epitope tag (DYKDDDDK) mouse monoclonal antibody conjugated with Alexa Fluor 647(Clone FLA-1, MBL lifescience, cat. no. M185-A64) |

| Validation | All the commercial antibodies were verified by the manufactures according to immunoblots and images on their websites. These were validated by their respective manufacturers as indicated at these links: https://labchem-wako.fujifilm.com/jp/product/detail/W01W0101-2238.html https://www.thermofisher.com/antibody/product/Goat-anti-Mouse-IgG-H-L-Highly-Cross-Adsorbed-Secondary-Antibody-Polyclonal/A32723 https://www.mblintl.com/products/m185-a64/ |
|---|---|

# Eukaryotic cell lines

Policy information about cell lines and Sex and Gender in Research

| Cell line source(s) | HEK293A purchased from Thermo Fisher Scientific (cat. no. R70507); HEK293S GnT1− purchased from American Type Culture Collection (cat. no. CRL-3022); E. Coli BL21 strain (DE3) purchased from New England Biolabs (cat. no. Cat# C2527), Sf9 (Gibco, cat. no. 11496015) |
|---|---|
| Authentication | None were authenticated. |
| Mycoplasma contamination | Not tested in Sf9; HEK 293A and HEK293S cells were regularly screened to ensure the absence of mycoplasma contamination using MycoAlert Mycoplasma detection kit (Lonza) |
| Commonly misidentified lines (See ICLAC register) | None of commonly misidentified lines were used in this study. |

# Flow Cytometry

## Plots

Confirm that:

☐ The axis labels state the marker and fluorochrome used (e.g. CD4-FITC).

☐ The axis scales are clearly visible. Include numbers along axes only for bottom left plot of group (a 'group' is an analysis of identical markers).

☐ All plots are contour plots with outliers or pseudocolor plots.

☐ A numerical value for number of cells or percentage (with statistics) is provided.

## Methodology

| Sample preparation | HEK293A cells were transfected by combining 3 μL of polyethylenimine solution (1mg mL-1) and 200 ng of a plasmid encoding FLAG epitope-tagged GPCR. |
|---|---|
| Instrument | EC800 flow cytometer (sony) |
| Software | FlowJo 10 (FlowJo), Prism 9 (GraphPad) |
| Cell population abundance | N/A |
| Gating strategy | Live cells were gated with a forward scatter (FS-Peak-Lin) cutoff of 390 setting a gain value of 1.7. |

☐ Tick this box to confirm that a figure exemplifying the gating strategy is provided in the Supplementary Information.

