## [Peer Review File · Nature]

Manuscript Title: Class-B1 GPCR activation by an intracellular agonist

Reviewer Comments & Author Rebuttals

Reviewer Reports on the Initial Version:

Referee expertise:

Referee #1: class B receptors, signalling

Referee #2: GPCRs, structure, cryo-EM

Referees' comments:

Referee #1 (Remarks to the Author):

This is an interesting manuscript that identifies a novel mechanism for an allosteric agonist of the PTH1R in a cryo-EM structure of the human PTH1R–Gs complex bound to a PTH1R agonist, PCO371, at 2.9 Å resolution. PCO371 binds with an unidentified intracellular pocket of PTH1R and directly interacts with Gs. They found that the PCO371-binding mode rearranges the intracellular region toward the active conformation without extracellularly induced allosteric signal propagation. They argue that PCO371 stabilizes the outwardly-bent conformation of TM6, facilitating binding to G-proteins rather than β -arrestins. Importantly, they find that the PCO371-binding pocket is highly conserved in class B1 GPCRs, although the mechanism underlying this conservation is not entirely clear.

I think the major novelty/finding in this paper is the mechanism of PCO371 acting as a “molecular wedge” of TM6, resulting in a common activation mechanism across a number of class B1 GPCRs. But I do have concerns about whether this signaling is: (1) truly biased; and (2) the underlying mechanism in the class B1 GPCRs. These will be raised below going through the data. I think a number of additional studies would be required to demonstrate the mechanism underlying PCO371's agonism and whether it is truly biased or just a weak partial agonist.

Major comments:

- Extended Figure 2: It appears that the structure was refined from the 111,962 particles (although it is noted as 109,422 in the Methods) that were 26.2% of the particles. Could the authors discuss the data from the 69.9% of particles (299,499 particles) that were not chosen for further refinement? What issues were there with that data and why wasn't that used for refinement?
- Figure 2e: There is a lot of discussion regarding the bend angle of TM6 that is induced by PCO371 (~145 degrees) compared to the more angular bend induced by PTH (~90-100 degrees). But it appears that the intracellular TM6 conformation is very similar between the two – the difference is largely driven by the fact that PCO371 does not induce any changes in the orthosteric site. So for all of these discussions on bend angle throughout the text, isn't it really just a lack of change in the

conformation of TM6 in the orthosteric site induced by PC0371 (so this is really just a question of semantics)?

- Figure 3f and Extended Data 7b: It is unclear as to whether some of these differences are truly significant (especially the EC50) as the vast majority of fits are based on a single measurement. By that I mean that the PTH1R estimate for EC50 is largely based on the cAMP level at 10-11 – all of the other values are either close to zero or to Emax. So the fit is really based on 1 data point. In this situation, it would be better to obtain two or three data points per log, especially if less than a log shift change in EC50 is being argued as being statistically significant. The G418L and P415L data is pretty obvious, but the other differences are minor. However, it is unclear why G418L Emax decrease with PTH is not statistically significant.

- Figure 3g: Why was the b1AR-Gs structure chosen as a comparison? I am not aware of any class B1 structures with arrestin, but could a model be developed? And is this same steric clash present in other arrestin structures.

- Figure 3h and Extended Figure 7a: This is my most significant concern. It is unclear whether there is true bias with PC0371 or whether it is just a very weak partial agonist. As can be seen from Ext Fig 7a, there is a -5 shift in cAMP between PTH and PC0371 (and this is an underestimate since PC0371 doesn't hit Emax). So for the barrestin assay with an EC50 of -9, a similar shift would result in an EC50 of -4, although likely worse based on the discussion above. So it is not surprising to see a lack of activity. While a bias plot or operational model approach (which would require a determination of the affinity of PC0371 for the receptor) could be used to analyze the data, I think it would still be very difficult to differentiate between weak partial agonism and true bias. A better barrestin assay is required – probably the most sensitive is the TANGO assay (although it can be prone to artifacts due to its modifications).

- Figure 3h. Related to that re: the NanoBiT assay for arrestin recruitment. This requires a modification of the receptor so could potentially be prone to artifact. An orthogonal assay (e.g., BRET, Tango, confocal) should be performed to confirm this finding. But if it is TANGO, it would require additional confirmation with single high-dose confocal microscopy.

- Figure 4: Other significant concern. The effect of PC0371 does not appear to be artifactual as it is absent at many receptors. And the L370P mutation of PTH2R that restores the PC0371 effect is very nice. But it is very unclear why GIPR (which shares the same sequence as GCGR in the region identified as critical for PC0371) or the other clade of B1 receptors does not have a response to PC0371 (as it would not be due to the 6x49 position). Determining this would greatly strengthen the manuscript, because as it stands, it is not clear what is responsible for the specificity of the PC0371 effect on only a subset of class B1 receptors.

- Related to this, it would significantly increase the impact of this study if there were any other pockets that could be exploited to generate bitopic ligands (or other approaches) to allow this mechanism to be exploited to yield receptor-specific agonists. The strength of allosteric modulators is their receptor specificity, but PC0371 does not have any such specificity (although the mechanism is interesting).

- Figure 4: Do the other receptors also demonstrate the same pH dependence as PTH1R?

Minor comments:

- Figure 2e. It would be helpful to have labels above each graph so referring back to the Figure legend is not required.

- Figure 3f – same issue – needs a figure legend.

- Line 225: “severely” should be “severe”

Referee #2 (Remarks to the Author):

This is a well-written and clearly illustrated manuscript describing the mechanism of a novel, completely G protein biased, non-peptide agonist (PCO371) for the PTHR1. PCO371 was previously thought to occupy the transmembrane domain orthosteric pocket because it was competitive in a peptide binding assay. The cryoEM structure reveals a novel binding site that is conserved in closely related Family B1 GPCRs that stabilizes the outward displacement of TM6 from the cytoplasmic side and also interacts with the end of the alpha 5 helix of Gs. The mechanism elucidated by the structure is supported by mutagenesis data. While it's not clear that this will be a druggable pocket for other GPCR families, I believe it will be of interest to the wider GPCR field.

As the authors note, PCO371 activates 7 Family B1 GPCRs, and suggest that “PCO371 is a potential seed for the development of small molecule-based drugs for class B1 GPCRs”; however, PCO371 suffers from a lack of selectivity. Perhaps the authors could speculate on ways one could enhance selectivity given that the amino acids that form the PCO371 binding pocket are highly conserved.

Minor point. Would suggest more contrasting colors than green and cyan for illustrations in Fig. 2 and 3.

Author Rebuttals to Initial Comments:

We are grateful to the Referees and the Editorial Board for critically reviewing our manuscript entitled “A novel activation mechanism of class B1 GPCRs via a conserved intracellular pocket” by Kobayashi *et al.* We have revised our manuscript by taking all of the Referees’ comments and concerns into account. Below are the detailed authors’ responses (in blue) to the Referees’ comments (in black).

Referees' comments:

Referee #1 (Remarks to the Author):

This is an interesting manuscript that identifies a novel mechanism for an allosteric agonist of the PTH1R in a cryo-EM structure of the human PTH1R–Gs complex bound to a PTH1R agonist, PCO371, at 2.9 Å resolution. PCO371 binds with an unidentified intracellular pocket of PTH1R and directly interacts with Gs. They found that the PCO371-binding mode rearranges the intracellular region toward the active conformation without extracellularly induced allosteric signal propagation. They argue that PCO371 stabilizes the outwardly-bent conformation of TM6, facilitating binding to G-proteins rather than β -arrestins. Importantly, they find that the PCO371-binding pocket is highly conserved in class B1 GPCRs, although the mechanism underlying this conservation is not entirely clear.

I think the major novelty/finding in this paper is the mechanism of PCO371 acting as a “molecular wedge” of TM6, resulting in a common activation mechanism across a number of class B1 GPCRs. But I do have concerns about whether this signaling is: (1) truly biased; and (2) the underlying mechanism in the class B1 GPCRs. These will be raised below going through the data. I think a number of additional studies would be required to demonstrate the mechanism underlying PCO371's agonism and whether it is truly biased or just a weak partial agonist.

Authors' response #1:

We appreciate your positive evaluation of our manuscript.

For the former concern (1), first, we performed additional β -arrestin recruitment assays with native PTH1R by the bystander method (PMID: 27397672) (**Response figure 5c, d**) by using native PTH1R, CAAX-fused LgBiT, and SmBiT fused β -arrestins. This experiment monitored β -arrestin recruitment without experimental artifacts and represented weak β -arrestin recruitment, compared to the situation where PCO371 would be a balanced agonist. To further demonstrate that PCO371 is a G-protein-biased agonist, we observed the co-localization of PTH1R and β -arrestin by confocal microscopy (**Response figure 6**). In brief, we expressed the FLAG tag fused PTH1R and mVenus-fused β -arrestin2 in HEK293 cells. After one day, the cells were incubated with an Alexa-647-labelled Flag-M1 antibody for one hour, and their membrane and subcellular localization were visualized using confocal microscopy (**Response Figure 6**). We observed the intracellular co-localization of PTH1R and β -arrestin at 30 minutes after high (1 μ M) and even low (10 pM) concentrations of PTH stimulation. In contrast, this intracellular co-localization was not observed with a high concentration (100 μ M) of PCO371 stimulation, which is equivalent to 10 pM PTH stimulation in the cAMP assay (**Response Figure 6**). Taken together, we conclude that PCO371 is a G protein-biased agonist.

For the latter concern (2), we performed additional glo-sensor cAMP accumulation assays with the four chimeric receptors and a computational analysis of PCO371-bound PTH1R. The three chimeric receptors (PTH1R-TM6 replaced by that of PAC1R or GLP-1R, and GCGR-TM6 replaced by that of GLP-1R) revealed that TM6 is a determinant for the PCO371 response as we described in the initial manuscript (**Response figure**

7). However, GCGR-TM6 replaced by GIPR impaired the response to endogenous glucagon; thus, a more precise comparison between GCGR and GIPR is difficult. To distinguish these two receptors, we carefully examined previously reported active class B1 GPCRs and noticed that asparagine (N)^{5.50} adopted alternative conformations between GCGR and GIPR. Moreover, the PCO371-insensitive receptors adopted a uniform N^{5.50} conformation toward TM3, while the PCO371-sensitive receptors altered the N^{5.50} conformation toward the center of the receptor (**Response figures 8a-c**). To determine what these distinct N^{5.50} conformations represent, we performed three independent molecular dynamics simulations, starting from our PCO371-PTH1R-Gs cryo-EM structure with the Gs protein removed. Our simulations showed that the N^{5.50} conformation is stable in its position in our PCO371-bound PTH1R cryo-EM structure and is distinct from those in PCO371-insensitive receptors, including GLP-1R. Notably, PCO371 created a new interaction with N374^{5.50} (**Response figures 8d [the bottom panel], 8g**), which is consistent with the fact that the N374^{5.50}A PTH1R mutant selectively reduced PCO371-induced receptor activation (**Response figure 8h**). These results indicate that the conformation of N374^{5.50} is crucial for PCO371 recognition, and N374^{5.50} can respond to PCO371 when the side chain of N374^{5.50} is oriented toward the center of the receptor (**Response figures 8c**). These computational and functional analyses indicate that N^{5.50} is one of the determinants for PCO371 recognition and PCO371 sensitivity in class B1 GPCRs, in addition to TM6 motility. Thus, the distinct conformation of N^{5.50} may explain the different responses of GCGR and GIPR, and a further PCO371-bound GCGR structure may provide a more comprehensive understanding of the receptor selectivity of PCO371.

Major comments:

- Extended Figure 2: It appears that the structure was refined from the 111,962 particles (although it is noted as 109,422 in the Methods) that were 26.2% of the particles. Could the authors discuss the data from the 69.9% of particles (299,499 particles) that were not chosen for further refinement? What issues were there with that data and why wasn't that used for refinement?

Authors' response #2:

Your suggestion is reasonable, and we thank you for pointing this out. We would like to explain why we selected only the minor population comprising 111,962 particles in the second 3D classification (**Response figures 1 and 2**).

First, we performed Laplacian-of-Gaussian picking and roughly extracted initial particles from the micrographs (2,942 micrographs, 2,591,640 particles). Subsequently, we eliminated some noisy or contaminated particles through 2D and 3D classifications. We then obtained a high-resolution potential map derived from 427,727 particles (**Response figure 1**). This initial map showed the identical binding mode of PCO371, and the overall structure is similar to the selected map (111,962 particles), as shown in the Response figure 1 below. In the initial map, the intracellular region of the receptor, together with the Gs protein, showed a homogenous conformation and the highest resolution in this complex. In contrast, the extracellular region showed poor density and high flexibility of the nature found in class B1 GPCRs, representing relatively low resolution (**Response figure 1**). To refine the quality of this map, we applied further classification to the 427,727 particles and distinguished 111,962 particles and 299,499 particles mainly based on the conformational heterogeneity in the extracellular region.

The map comprising 299,499 particles still showed a conformational heterogeneity in the extracellular region, representing a relatively low-resolution map than the selected map (**Response figure 1**). In contrast, the selected map comprising 111,962 particles represented an unambiguous density and higher global resolution

than the other maps (**Response figure 1**). Thus, we selected the map comprising 111,962 particles and performed further classification.

Response figure 1 Classification strategy in cryo-EM workflow

Local resolution map and TMD map between the initial (comprising 427,727 particles), non-selected (comprising 299,499 particles), and the selected maps (comprising 109,422 particles).

In the subsequent procedure, the no-align classification panel was missing from Extended Data Figure 2. We thank you for indicating its absence and added it in Extended Data Figure 2. To further refine the resolution, we applied the no-align classification to the 111,962 particles and isolated the final particles (109,422 particles) as shown in the Response figure 2.

Through this process, we obtained the clear density map, and the number of particles described in the Methods section (page54, line 924) is correct.

Extended Data Figure 2

The refined cryo-EM data processing flowchart. We added the third 3D classification figure that was missing from this figure.

- Figure 2e: There is a lot of discussion regarding the bend angle of TM6 that is induced by PC0371 (~ 145 degrees) compared to the more angular bend induced by PTH (~90-100 degrees). But it appears that the intracellular TM6 conformation is very similar between the two – the difference is largely driven by the fact that PC0371 does not induce any changes in the orthosteric site. So for all of these discussions on bend angle throughout the text, isn't it really just a lack of change in the conformation of TM6 in the orthosteric site induced by PC0371 (so this is really just a question of semantics)?

Authors' response #3:

Thank you for mentioning this. As the Referee suggested, the most notable difference is in the extracellular region than the intracellular region, compared to the other active class B1 GPCR structures. It should be noted that the other active structures showed the identical sharp kink of TM6, indicating the common structural transition for an active structure from the extracellular side to the intracellular side. Comparing TM6 of our structure with that of active and inactive structure highlights PCO371 doesn't induce this structural transition, which may be a criteria for other intracellular agonists and biased signaling mechanisms in the future. Thus, we believe that the TM6 comparison is useful for readers.

To emphasize the notable difference in the TM6 conformation, we revised our manuscript as described below. (Page 10, lines 187-191)

This unique extracellular conformation caused TM6 to unwind and become moderately kinked (approximately 145°) in the PCO371-bound PTH1R structure, which is distinct from the sharply kinked TM6 (approximately 90°) in the PTH-bound PTH1R structure (Extended Data Fig. 5b).

- Figure 3f and Extended Data 7b: It is unclear as to whether some of these differences are truly significant (especially the EC50) as the vast majority of fits are based on a single measurement. By that I mean that the PTH1R estimate for EC50 is largely based on the cAMP level at 10-11 – all of the other values are either close to zero or to Emax. So the fit is really based on 1 data point. In this situation, it would be better to obtain two or three data points per log, especially if less than a log shift change in EC50 is being argued as being statistically significant. The G418L and P415L data is pretty obvious, but the other differences are minor. However, it is unclear why G418L Emax decrease with PTH is not statistically significant.

Authors' response #4:

According to the Referee's suggestion, we repeated the functional experiments by doubling the number of data points (i.e., two data points per log). (Response Figures 3). Indeed, the additional data points improved the smooth fitting of the sigmoidal curve by having more data points around the inflection point. Both the potency and efficacy values in this experiment differed by less than 2-fold from those in the previous experiment (Response Figure 3). In the revised manuscript, we replaced the figure with the newly obtained data (Extended Data Figure 7). In this revision, we also applied this method for obtaining data points in subsequent additional experiments. (Please see also Response Figures 5, 7, and 10)

In terms of the statistical concern about the use of one-way ANOVA and post hoc Dunnett's test for each of the potency and efficacy panels, the actual p-values are shown in Supplementary Table 1. This statistical group-analysis method is generally used to evaluate the statistical significance of three or more elements (i.e., one wild type and two or more mutants) compared to one reference (the wild type). It is important to note that this method considers the values of all mutants for the calculation of a single p-value and, therefore, may not show statistical significance for certain comparisons, even if there is an apparent difference between one mutant and the wild type.

EC50	PTH(1-34)	Adjusted P Value
	WT vs. P415L	0.1396
	WT vs. L416A	<0.0001
	WT vs. F417A	<0.0001
	WT vs. G418L	<0.0001
	WT vs. Q451A	<0.0001
EC50	PCO371	Adjusted P Value
	WT vs. P415L	<0.0001
	WT vs. L416A	0.0191
	WT vs. F417A	<0.0001
	WT vs. G418L	0.4066
	WT vs. Q451A	0.8992
E _{max}	PTH(1-34)	Adjusted P Value
	WT vs. P415L	0.3108
	WT vs. L416A	0.775
	WT vs. F417A	0.9998
	WT vs. G418L	0.0596
	WT vs. Q451A	0.9746
E _{max}	PCO371	Adjusted P Value
	WT vs. P415L	<0.0001
	WT vs. L416A	0.2984
	WT vs. F417A	0.989
	WT vs. G418L	0.2339
	WT vs. Q451A	0.6462

Supplemental Table 1

ANOVA and post hoc Dunnett's test for each of the potency and efficacy panels, the actual p-values

Response figure 3 Revised cAMP accumulation assay

Finely sampled GloSensor cAMP responses upon PTH and PCO371 stimulation. As described in the revised manuscript, these cAMP accumulations were detected by doubling the number of data points from the previous experiments.

- Figure 3g: Why was the b1AR-Gs structure chosen as a comparison? I am not aware of any class B1 structures with arrestin, but could a model be developed? And is this same steric clash present in other arrestin structures.

Authors' response #5:

As Referee 1 indicated, no structures of class B1 GPCR- β -arrestin complexes have been reported yet, and the structures of GPCR- β -arrestin complexes were reported for class A GPCRs. A comparison of PTH1R and class A GPCR- β -arrestin complex structures revealed that β 1AR is the most adequate, as it mainly binds Gs and β -arrestin in similar manners to PTH1R. In addition, the β 1AR-Gs and β 1AR- β -arrestin structures showed differential binding modes for each transducer, leading to preferential signal transduction. Thus, we compared our PTH1R structure to β 1AR-Gs and β 1AR- β -arrestin among several GPCR- β -arrestin complexes.

As Referee 1 suggested, we generated a predicted model with PTH1R- β -arrestin complexes by using complex structure prediction by AlphaFold2. AlphaFold2 successfully generated a complex that is not separated structures of a PTH1R and β -arrestin. Superimposition of the predicted model and the β 1AR- β -arrestin structure showed that the β -arrestin orientation in the predicted model is similar to that in the β 1AR- β -arrestin structure. Moreover, the superimposition of our PCO371-PTH1R-Gs cryo-EM structure and the predicted model revealed that β -arrestin severely clashes with PCO371 (**Response Figure 4a, b**). Thus, the steric clash of β -arrestin is also important for biased signaling by PCO371.

Nevertheless, we wondered if a predicted model would be useful as a class B1 GPCR- β -arrestin complex for two reasons. First, the predicted structure showed sharply kinked TM6 conformation, which is a significant hallmark of active class B1 GPCR-Gs complexes. This TMD structure indicated that the predicted TMD conformation is strongly influenced by the class B1 GPCR-Gs structures deposited in the PDB, as AlphaFold2 refers to these deposited PDB structures. Second, the predicted GPCR- β -arrestin complexes seem to be strongly related to the class A GPCR- β -arrestin complexes at present, as most of the structure prediction programs, including AlphaFold2, use the feature from a similar deposited structure for highly accurate structural prediction. For these reasons, although Referee 1's suggestion is very reasonable, we believe that a direct comparison of our structure and the experimental and highly accurate β 1AR complexes will provide the clearest information.

Response figure 4 Structural comparison with a predicted PTH1R- β -arrestin complex

a, b Docking structure of PTH1R- β -arrestin (from β 1AR) and predicted model of PTH1R- β -arrestin complex (left and center panels). These structures showed similar binding modes.

c, Predicted Local Distance Difference Test (pLDDT) score. The predicted complex is colored based on the pLDDT score by the gradient color, indicating that the TM6 and β -arrestin interface is poorly predicted because no similar complex structure has been deposited in the PDB.

- Figure 3h and Extended Figure 7a: This is my most significant concern. It is unclear whether there is true bias with PC0371 or whether it is just a very weak partial agonist. As can be seen from Ext Fig 7a, there is a -5 shift in cAMP between PTH and PC0371 (and this is an underestimate since PC0371 doesn't hit Emax). So for the barrestin assay with an EC50 of -9, a similar shift would result in an EC50 of -4, although likely worse based on the discussion above. So it is not surprising to see a lack of activity. While a bias plot or operational model approach (which would require a determination of the affinity of PC0371 for the receptor) could be used to analyze the data, I think it would still be very difficult to differentiate between weak partial agonism and true bias. A better barrestin assay is required – probably the most sensitive is the TANGO assay (although it can be prone to artifacts due to its modifications).

- Figure 3h. Related to that re: the NanoBiT assay for arrestin recruitment. This requires a modification of the receptor so could potentially be prone to artifact. An orthogonal assay (e.g., BRET, Tango, confocal) should be performed to confirm this finding. But if it is TANGO, it would require additional confirmation with single high-dose confocal microscopy.

Authors' response #6:

Thank you for the thoughtful suggestion. We re-calculated our initial data and performed two additional experiments with a native PTH1R.

First, to evaluate the signal bias of PCO371, we attempted to obtain its signal-bias factor from the data in the initial manuscript by using methods (a) to (c), as below.

(a) Bias plot

We generated a bias plot based on the previous reports (PMID 35106752, 28290478). (Response Figure 5a) The signal bias could not be visualized using this method due to the significant difference in the sensitivities of the two assays and the weak activity of PCO371.

(b) Operational model

As pointed out, the K_d value of PCO371 needs to be determined for applying this model. We synthesized ^{14}C -labelled PCO371 and attempted to determine its K_d value, but were unable to obtain the value; therefore, the calculation of the bias factor in this method was not feasible.

(c) $\Delta\Delta\log$ (Emax/EC50) method

We calculated the bias factor of PCO371 using the $\Delta\Delta\log$ (Emax/EC50) method (PMID 35106752, 18253722). Since the β -arrestin recruitment response in PCO371 was completely undetectable, the theoretical β -arrestin recruitment response, hypothetically regarding when PCO371 as a balanced agonist, was calculated by considering the Gs/ β arr activity of PTH and the Gs activity of PCO371 and compared with the actual β -arrestin recruitment response of PCO371 (Response Figure 5b). The actual β -arrestin recruitment response curves of PCO371 are lower than the theoretical curves, indicating that PCO371 is a G protein-biased agonist for PTH1R.

Subsequently, we additionally evaluated β -arrestin recruitment to native PTH1R to examine the Referee's concerns about experimental artifacts. Because the above β -arrestin recruitment assay uses a construct with a luciferase fragment fused to PTH1R, we also evaluated the β arr1/2 recruitment activity by the bystander

method (PMID: 27397672), as another approach, wherein a plasma membrane localization sequence (CAAX) is fused to LgBiT (**Response Figure 5c**). In this method, β -arrestin recruitment to the plasma membrane can be evaluated without the need to modify the receptor. In addition, we attempted to further increase the sensitivity of the β -arrestin assay by combining the overexpression of GRK2 and a C-term truncated (Δ C382) pre-active mutant of β -arrestin.

We observed slight PCO371-induced β -arrestin recruitments (4.4-6.2% of Emax of PTH) by overexpressing the GRK2 or a C-term truncated β -arrestin, but they were much smaller than the response when PCO371 is hypothesized to be a balanced ligand (**Response Figure 5d, e**). We used untagged, native PTH1R in Response Figure 5, in which we measured the β -arrestin recruitment response by the bystander method, also demonstrating that PCO371 is a G protein-biased agonist.

Finally, to thoroughly demonstrate that PCO371 is a G-protein-biased agonist, we assessed the intracellular translocation of PTH1R/ β -arrestin by using confocal microscopy. We expressed FLAG tag fused PTH1R and mVenus-fused β -arrestin2 in HEK293 cells. After one day, the cells were incubated with an Alexa-647-labelled Flag-M1 antibody for one hour, and their membrane and subcellular localizations were visualized using confocal microscopy (**Response Fig. 6**). After 30 min, we observed the intracellular co-localization of PTH1R and β -arrestin2 at a high concentration (100 nM) and even a low concentration (10 pM) of PTH stimulation. In contrast, this intracellular co-localization was not observed upon high concentration (100 μ M) PCO371 stimulation, which is equivalent to 10 pM PTH stimulation in the cAMP assay. Taken together, we conclude that PCO is a G protein-biased agonist.

Response figure 5 Additional functional studies

a, Bias plot derived from the GloSensor cAMP response (horizontal axis) and β -arrestin recruitment response (vertical axis) upon PTH and PCO371 stimulation.

b, Concentration-response curves of the β -arrestin1/2 recruitment response. The red dashed line indicates the theoretical β -arrestin recruitment response when PCO371 is hypothesized to be a balanced agonist. The solid line is the same as Fig. 3h.

c, Graphic representation of the NanoBiT-based bystander β -arrestin recruitment assay.

d, e, Concentration-response curves of β -arrestin1 (d) and β -arrestin2 recruitment response (e).

Response Figure 6 Intracellular co-localization of PTH1R and β arr2

a, b, Representative images of the cellular localization of Alexa-647-labeled PTH1R (magenta) and mVenus-fused β arr2 (green).

c, Quantification of co-localization of PTH1R and β arr2 upon vehicle, 100 nM PTH, 10 pM PTH, and 100 μ M PCO371 stimulation. The colocalization index for individual cells in each stimulation condition was calculated by Fiji (ImageJ). Symbols and error bars represent means and SEM, respectively, of 10-19 cells. * and ** represent $P < 0.05$ and 0.01 , respectively, with one-way ANOVA followed by the Dunnett's test for multiple comparison analysis with reference to the vehicle stimulation.

- Figure 4: Other significant concern. The effect of PCO371 does not appear to be artifactual as it is absent at many receptors. And the L370P mutation of PTH2R that restores the PCO371 effect is very nice. But it is very unclear why GIPR (which shares the same sequence as GCGR in the region identified as critical for PCO371) or the other clade of B1 receptors does not have a response to PCO371 (as it would not be due to the 6x49 position). Determining this would greatly strengthen the manuscript, because as it stands, it is not clear what is responsible for the specificity of the PCO371 effect on only a subset of class B1 receptors.

Authors' response #7

To further describe the receptor selectivity of PCO371, we additionally performed mutant analyses for additional mutants and three independent MD simulation runs.

First, we mentioned that the selectivity of PCO371 may depend on the different motility of TM6 in the initial manuscript. In brief, structural studies of GLP-1R revealed three different ligand binding modes: orthosteric endogenous agonist, orthosteric engineered chemical agonist, and intracellular outer helical engineered agonist, as shown in Extended Data Fig.9b; also, please see below (**Response Figure 7a**). All the structures represented similar TM6 kinks, even though the intracellular agonist-bound structure has no agonists in the orthosteric pocket, as shown in Extended Data Fig.9c; please see below (**Response Figure 7a**). These observations suggest that the extracellular and intracellular halves of the TM6 movement of GLP-1R are closely

linked and cannot accommodate PCO371 due to the conformational clash of TM6. Thus, we hypothesized that PCO371 only activates a group of GPCRs that do not require the inside-out change for their activation, since PCO371 is unable to bind to receptors that adopt the typically activated extracellular conformation (**Response Figure 7b**). To prove this hypothesis, we designed three chimeric receptors, PTH1R (TM6 replaced by that of GLP-1R), PTH1R (TM6 replaced by that of PAC1R), GCGR (TM6 replaced by that of GLP-1R), and GCGR (TM6 replaced by that of GIPR), and measured cAMP production by PCO371 (**Response Figure 7c, d**). Interestingly, only minor differences in their endogenous ligand response were detected, except for GCGR (TM6 replaced by that of GIPR), suggesting these three chimeric receptors retain the receptor activity. Notably, PTH1R (TM6-GLP-1R) completely abolished cAMP production by PCO371, while PTH1R (TM6-PAC1R) showed distinguishable cAMP production by PCO371. In line with our notion, GCGR (TM6-GLP-1R) also mostly lacked cAMP production. These results strongly support our idea that PCO371 activates the receptors that can independently mobilize the intracellular half of TM6 from the extracellular half. However, GCGR (TM6-GIPR) altered the endogenous glucagon response, suggesting that this mutant receptor may not keep its inherent functionality in GCGR. Thus, although this mutant also slightly decreases the PCO371-response, a direct comparison of GCGR and GIPR is difficult. For this unique ability of GCGR, further structural analysis of PCO371-bound GCGR may reveal the comprehensive receptor selectivity of PCO371. We added discussions of these findings, including WT PTH and GCGR and mutant PTH1R (TM6-GLP-1R), PTH1R (TM6-PAC1R), and GCGR (TM6-GLP-1R), in Results section (page 18, line 297-307) in the manuscript and panels in Extended Data Figure 9d.

Subsequently, to distinguish the PCO371-sensitivities between GCGR and GIPR, we superimposed and carefully inspected previously reported endogenous agonist-bound active class B1 GPCRs. We noticed an alternative conformation of N^{5.50} between the two receptors (**Response figure 8a–b**). The superimposed active class B1 GPCR structures demonstrated that the PCO371-insensitive receptors (please see also main Fig. 4d and **Response figure 7c**) adopt the outward N^{5.50} conformation toward TM3, while the PCO371-sensitive receptors adopt the inward conformation toward the center of the receptor (**Response figure 8c**). Then, we performed three independent molecular dynamics simulations, starting from our PCO371-PTH1R-Gs cryo-EM structure with the Gs protein removed, and analyzed the N^{5.50} functionality for PCO371 recognition. These simulations showed that the TMD of the receptor maintained an active conformation similar to that in the cryo-EM structure, and PCO371 stably bound to the receptor (**Response figure 8d [top 2 panels]**). The N^{5.50} conformation is stable in the position of our PCO371-bound PTH1R cryo-EM structure and is distinct from that in the active GLP-1-bound GLP-1R structure (PDB: 6X18) (**Response figure 8d [second from the bottom panel] and 8e**). Notably, in run 2, the receptor showed an intermediate state, and PCO371 created a new and stable interaction with N374^{5.50} (**Response figure 8d [the bottom panel], 8f, and 8g**). Consistent with the simulation, the N374^{5.50}A PTH1R mutant selectively reduced PCO371-induced receptor activation, while this mutant had no effect on PTH-induced receptor activation (**Response figure 8h**). These results suggest that the side-chain orientation of N374^{5.50} is crucial for PCO371 recognition, and N374^{5.50} can respond to PCO371 when its side chain is oriented toward TM6 (**Response figure 8c**). Combined with our MD simulation and functional study, our additional results suggest that N^{5.50} is one of the determinants for PCO371 recognition and PCO371 sensitivity in class B1 GPCRs, in addition to the distinct motility of TM6. Thus, the distinct conformations of N^{5.50} may explain the different responses to GCGR and GIPR, and a PCO371-bound GCGR structure may provide a more comprehensive understanding for the receptor selectivity of PCO371. We added these findings in the discussion section (page 20-22, line 341-367) in the manuscript and panels in Extended Data Figure 10.

Response
figure 7
cAMP

Extended Data Fig.9b and 9c

Main fig. 2d

Red: PCO371-sensitive receptors

Blue: PCO371-insensitive receptors

Main fig. 4d

accumulation with TM6-replaced mutants

(a, b) Reference of main Figs.2d, 4d, and Extended Data Fig.9b, c.

(c) PTH and PCO371 induced a cAMP response via the WT and mutant receptors. The bottom panel depicts PCO371-sensitive and -insensitive receptors as shown in main Fig. 4d.

Response figure 8 Computational and functional analyses of PCO371-bound PTH1R

MD simulation with PCO371-bound PTH1R.

a, A cross-section view of the PCO371-bound PTH1R.

b, A magnified view of the superimposed structures of PCO371-bound PTH1R (purple), GCG-bound GCGR (orange), and GIP-bound GIPR (gray).

c, Structural comparison of N5.50 of PCO371-insensitive receptors and PCO371-sensitive receptors, related to Fig.4d. Purple, PCO371-bound PTH1R; gray, PCO371-insensitive receptors; orange, PCO371-sensitive receptors.

d, Trajectory analysis of three independent runs with PCO371-bound PTH1R. The top two panels show that PCO371-bound PTH1R maintains an active conformation, as visualized in our cryo-EM structure. The second from the bottom panel shows the dihedral angle of N374^{5.50}, revealing that N374^{5.50} in PTH1R is located toward TM6 in the whole simulation, contrast to that of the active GLP-1R (PDB ID: 6X18). The bottom panel shows the distance between N374^{5.50}-ND2 and PCO371-O1, representing that PCO371 stably interacts with N374^{5.50} in an intermediate state.

e, Distinct conformations of N374^{5.50} between PCO371-bound PTH1R and GLP-1-bound active GLP-1R (PDB ID: 6X18). The dihedral angle is calculated with the ω and ψ angles of N^{5.50} (N-C α and C β -C γ).

f, Superimposition of our cryo-EM structure (purple) and a representative snapshot of the 1.5 μ s MD simulation in run 2 of PCO371-bound PTH1R (yellow). In the simulation, the receptor adopted an intermediate structure, and PCO371 moved toward TM5.

g, Structural comparison of our cryo-EM structure (purple) and an intermediate structure (yellow) observed in the simulation. PCO371 does not interact with N374^{5.50} in our cryo-EM structure, while it creates a stable interaction with N374^{5.50} in the simulation.

h, cAMP accumulation of WT and N374^{5.50}A mutant with PTH and PCO371. The N374^{5.50}A mutant selectively reduced the PCO371 response.

- Related to this, it would significantly increase the impact of this study if there were any other pockets that could be exploited to generate bitopic ligands (or other approaches) to allow this mechanism to be exploited to yield receptor-specific agonists. The strength of allosteric modulators is their receptor specificity, but PCO371 does not have any such specificity (although the mechanism is interesting).

Authors' response #8

Thank you for the suggestion. Accordingly, we discuss and propose a way to gain receptor selectivity and develop a bitopic ligand.

In general, extracellular agonists gain receptor selectivity by elongating the ligand toward the outside of the receptor because outside of the receptor shows sequential diversity ([https://www.cell.com/molecular-cell/pdfExtended/S1097-2765\(21\)00504-9](https://www.cell.com/molecular-cell/pdfExtended/S1097-2765(21)00504-9)). Our structure showed that the intracellular cavity of PCO371-bound PTH1R is tightly sealed by Gs, and PCO371 cannot interact with the intracellular loops (**Response figures 9a and 9b**). Instead, the ligand-binding pocket of PCO371 is laterally open between TM6 and TM7, which is accessible to Helix 8 (**Response figure 9b**).

Moreover, our structure revealed an intracellular pocket consisting of TM1, TM7, and Helix 8, facing toward the membrane region (**Response figure 9c [left]**). Given the sequence diversity in Helix 8 among class B1 GPCRs (**Response figure 9d**), a molecule bound to this pocket may provide receptor selectivity, leading to the development of a bitopic ligand. Furthermore, PTH1R possesses a non-conserved cysteine at the position 7.60 (**Response figure 9c [right]**). Note that PCO371 does not activate CTR and PTH2R, although these receptors contain the cysteine at the position 7.60. Thus, the addition of a covalent functional group to PCO371 may provide PCO371 receptor-selectivity toward PTH1R. C β of Cys^{7.60} is located 6.4 Å away from O6 of PCO371, and three or four carbon residues would be sufficient for connecting PCO371 and C^{7.60}. In addition, a previous covalent agent (Compound 2) for GLP-1R also may be useful for ligand generation (<https://www.nature.com/articles/s41467-021-24058-z>). Compound 2 acts as a weak covalent bond agent and

selectively binds C347^{6,36}. By optimizing compound 2 to fit cavity formed by TM6, TM7, and Gαs of PTH1R, a modified compound2-conjugated PCO371 may become a PTH1R selective agonist. We mention these points in the discussion section (page 22-23, lines 369-390) in the manuscript and Extended Data Figure 11.

Response figure 9 Putative drug development strategy for a bitopic agonist.

a, TMD structures of PCO371–PTH1R–Gs are shown parallel to the membrane.

b,c Cross-section and magnified views of TMD. The white dashed line shows an intracellular ligand binding pocket comprising TM1, TM7, and Helix 8.

d, Sequence alignment of the C-terminal region after TM7. The red box shows C462^{7,60} of PTH1R

- Figure 4: Do the other receptors also demonstrate the same pH dependence as PTH1R?

Authors' response #10:

We additionally performed the same experiment as in Fig. 4c and Extended Data Fig. 8d under pH 9.0 assay buffer conditions. (Response Figure 10). For two receptors (GHRHR, SCTR), there was an increase in efficacy under the pH 9.0 conditions, but no increased responses for the remaining four receptors (VIP1R, VIP2R, PAC1R, and GCGR) that respond to PCO371 under the pH 7.4 conditions. Importantly, the other eight receptors (PTH2R, GIPR, GLP1R, GLP2R, CRF1R, CRF2R, CALCR, and CALRL) do not respond under both pH 7.4 and pH 9.0 conditions.

Response Fig. 10 PCO371-induced activation of class B1 GPCRs under pH 9.0 buffer conditions.

Concentration-response curves of GloSensor cAMP responses of 15 class B1 GPCRs upon PCO371 (red) stimulation. Symbols and error bars represent the mean and SEM, respectively, of 5 independent experiments with each performed in duplicate.

Minor comments:

- Figure 2e. It would be helpful to have labels above each graph so referring back to the Figure legend is not required.
- Figure 3f – same issue – needs a figure legend.
- Line 225: “severely” should be “severe”

Authors' response #11

By following Referee#1's advice, we revised our manuscript as below.

1. In Fig. 2e, we removed the legends, and the GPCR structures were labeled near the plots.
2. In Fig. 3f, we added the legend above the panel.
3. In line 225 of the revised manuscript, we replaced the term “severely” with “severe”.

Referee #2 (Remarks to the Author):

This is a well-written and clearly illustrated manuscript describing the mechanism of a novel, completely G protein biased, non-peptide agonist (PCO371) for the PTHR1. PCO371 was previously thought to occupy the transmembrane domain orthosteric pocket because it was competitive in a peptide binding assay. The cryoEM structure reveals a novel binding site that is conserved in closely related Family B1 GPCRs that stabilizes the outward displacement of TM6 from the cytoplasmic side and also interacts with the end of the alpha 5 helix of Gs. The mechanism elucidated by the structure is supported by mutagenesis data. While it's not clear that this will be a druggable pocket for other GPCR families, I believe it will be of interest to the wider GPCR field.

As the authors note, PCO371 activates 7 Family B1 GPCRs, and suggest that "PCO371 is a potential seed for the development of small molecule-based drugs for class B1 GPCRs"; however, PCO371 suffers from a lack of selectivity. Perhaps the authors could speculate on ways one could enhance selectivity given that the amino acids that form the PCO371 binding pocket are highly conserved.

Authors' response #12

Thank you for an intriguing suggestion. We reviewed our structure and will propose a potential strategy for gaining receptor selectivity and generating a bitopic ligand. (Please also refer to **Authors' response #8**). In brief, PCO371 is accessible to the PCO371-binding pocket, and the conjugation of PCO371 and the putative binder would lead to the development of a bitopic ligand.

In general, extracellular agonists gain receptor selectivity by elongating the ligand toward the outside of the receptor because outside of the receptor shows sequential diversity ([https://www.cell.com/molecular-cell/pdfExtended/S1097-2765\(21\)00504-9](https://www.cell.com/molecular-cell/pdfExtended/S1097-2765(21)00504-9)). Our structure showed that the intracellular cavity of PCO371-bound PTH1R is tightly sealed by the $\alpha 5$ helix of Gs, and PCO371 cannot interact with the intracellular loops (**Response Fig. 9a and 9b**). Instead, the ligand-binding pocket of PCO371 is laterally open between TM6 and TM7, which is accessible to the outside of the receptor (**Response Fig. 9b**). PCO371-bound PTH1R displayed an intracellular ligand pocket formed by TM1, TM7, and Helix 8 (**Response Fig. 9c [left]**). Given that Helix 8 has a diverse sequence in class B1 GPCRs, this pocket has the potential for developing a receptor-specific binder (**Response Fig. 9d**).

Moreover, for a PTH1R selective PCO371 agonist, the addition of a covalent functional group to PCO371 may be useful. PTH1R possesses non-conserved cysteine in position 7.60, 6.4 Å apart from O6 of PCO371, which is only conserved in PTH1R among the PCO371-sensitive class B1 GPCRs (**Response Fig. 9c [right] and 9d**). Thus, an additional covalent functional group to PCO371 may increase the affinity for PTH1R and improve the receptor selectivity of PCO371.

Response figure 9 Putative drug development strategy for a bitopic agonist.

a, TMD structures of PCO371–PTH1R–Gs are shown parallel to the membrane.

b,c Cross-section and magnified views of TMD. The white dashed line shows an intracellular ligand binding pocket comprising TM1, TM7, and Helix 8.

d, Sequence alignment of the C-terminal region after TM7. The red box shows C462^{7.60} of PTH1R

Minor point. Would suggest more contrasting colors than green and cyan for illustrations in Fig. 2 and 3.

Authors' response #13

We reconsidered the clarity and uniformity of the colors throughout the figures and changed the colors of the PCO371-bound PTH1R from cyan to violet.

Reviewer Reports on the First Revision:

Referees' comments:

Referee #1 (Remarks to the Author):

The authors have done an excellent job responding to the reviewer suggestions. I do think one persistent limitation is that PC0371 is only a very weakly G protein-biased agonist at best – I think the authors have done about as thorough a job that can be performed to look at bias with this compound and the differences between a balanced agonist and what is observed with PC0371 is really quite minimal. That being said, the other findings in the manuscript should be of interest to a broad scientific audience and provide important insights into the signaling mechanisms of class B receptors.

Referee #2 (Remarks to the Author):

EF1, EF9D, and EF10h: Error bars should be defined in the legend.

Author Rebuttals to First Revision:

TITLE: Titles cannot exceed 75 characters (including spaces); they must not contain punctuation. We suggest "Parathyroid hormone receptor activation by an intracellular agonist".

Author's response #1

Thank you for examination. We revised title according to your suggestion.

SUMMARY PARAGRAPH: Papers start with a fully referenced, bold paragraph, ideally of about 200 words, aimed at readers in other disciplines. Numbers, abbreviations, acronyms or measurements should be avoided unless essential. The summary paragraph consists of 2 to 3 sentences of basic-level introduction to the field; a brief account of the background and rationale of the work; a statement of the main conclusions (introduced by the phrase 'Here we show' or its equivalent); and a conclusion of 2 to 3 sentences putting the main findings into general context so it is clear how the results described in the paper have moved the field forward. A downloadable, annotated example is available at <https://www.nature.com/nature/for-authors/formatting-guide>.

- Please trim the summary paragraph to within 230 words.

Author's response #2

Thank you for examination. We revised the abstract section with 230 words.

MAIN TEXT: If further introductory material is necessary, the main text can begin with up to 500 words of introduction expanding on the background to the work (some overlap with the summary is acceptable), before proceeding to a concise, focused account of the findings, and ending with 1 or 2 short paragraphs of discussion. Sections are separated with subheadings (up to 40 characters including spaces) to aid navigation.

Author's response #3

Thank you for examination. We have revised the introduction in 500 words.

- All the figures are quite large at the moment. Please ensure the final versions do not exceed 17 cm in height.

- I suggest moving Fig. 5 to the Extended Data (ED), seeing as this is just a representative cartoon. Remember that ED form part of the main paper online.

- I also propose moving panels e and f from Fig. 2 to ED. Along with a bit of reformatting this should also make the figure more manageable in size.

Author's response #4

Thank you for examination. We resized all figures within 180 mm x 170 mm square. To conduct this, we reorganized Figs. 2 and 3, and Extended Data Figs 5, and 6, as described below.

- Please draw PCO371 using our in-house style settings, also found on recent versions of ChemDraw. We will require a .cdxml file in addition to the required figure formats noted above. Please also remove the coloured boxes around parts of the molecule, as these detract from the message of the figure panel.

Previous version	Final version
Fig. 2e	EDF. 5c
Fig. 2f	EDF. 5d
Fig. 3a	-
Fig. 3b	EDF. 6b
Fig. 3c	Fig. 3a
Fig. 3d	Fig. 3b
Fig. 3e	Fig. 3c
Fig. 3f	Fig. 3d
Fig. 3g	Fig. 3e
Fig. 3h	Fig. 3f
Fig. 3i	Fig. 3g
EDF. 5c	-
EDF. 5d	Fig. 2e
EDF. 6b	EDF. 6c, d
EDF. 6c	EDF. 6e
EDF. 6d	EDF. 6f, g
EDF. 6e	EDF. 6h
EDF. 6f	EDF. 6i

Author's response #5

Thank you for this indication. We re-described the PCO371 chemical structure and provided .cdx file. In addition, we removed colored boxes.

- Please move the Cryo-EM reporting table to the ED.
- We can permit 12 ED items in this case (figures and tables combined).

Author's response #6

Thank you for these instructions. We rearranged the additional cryo-EM tables in the EDF. Nevertheless, we already included 11 items in EDF without the two cryo-EM tables, as we added additional materials in response to reviewer suggestions. We

would be grateful if you could accept 13 EDF items for a comprehensive revised paper.

Referees' comments:

Referee #1 (Remarks to the Author):

The authors have done an excellent job responding to the reviewer suggestions. I do think one persistent limitation is that PC0371 is only a very weakly G protein-biased agonist at best – I think the authors have done about as thorough a job that can be performed to look at bias with this compound and the differences between a balanced agonist and what is observed with PC0371 is really quite minimal. That being said, the other findings in the manuscript should be of interest to a broad scientific audience and provide important insights into the signaling mechanisms of class B receptors.

Author's response #7

We sincerely appreciate for the favorable response from Reviewers #1.

As indicated by the reviewer #1, PCO371 exerts relatively low affinity than most of the typical GPCR agonists. In line with the suggestion of reviewer #1, we have more precisely demonstrated the signal bias of PCO371 using confocal microscopy. We believe that this intriguingly biased mechanism is the pioneer for future ligand development, and modified PCO371 with more high affinity for PTH1R will provide comprehensive differences in the distinct signaling mode of PTH and PCO371.

Referee #2 (Remarks to the Author):

EF1, EF9D, and EF10h: Error bars should be defined in the legend.

Author's response #8

Thank you for suggestion. We added the description of symbols and error bars.